# Repetitive marsquakes in Martian upper mantle

Weijia Sun ⓘ [1] & Hrvoje Tkalčić ⓘ [2✉]

Marsquakes excite seismic wavefield, allowing the Martian interior structures to be probed. However, the Martian seismic data recorded by InSight have a low signal-to-noise ratio, making the identification of marsquakes challenging. Here we use the Matched Filter technique and Benford's Law to detect hitherto undetected events. Based on nine marsquake templates, we report 47 newly detected events, >90% of which are associated with the two high-quality events located beneath Cerberus Fossae. They occurred at all times of the Martian day, thus excluding the tidal modulation (e.g., Phobos) as their cause. We attribute the newly discovered, low-frequency, repetitive events to magma movement associated with volcanic activity in the upper mantle beneath Cerberus Fossae. The continuous seismicity suggests that Cerberus Fossae is seismically highly active and that the Martian mantle is mobile.

[1] Key Laboratory of Earth and Planetary Physics, Institute of Geology and Geophysics, Chinese Academy of Sciences, Beijing, China. [2] Research School of Earth Sciences, The Australian National University, Canberra, Australia. ✉email: hrvoje.tkalcic@anu.edu.au

Before NASA's InSight mission[1], Mars' size, mass, and moment of inertia provided initial estimates of its internal structure[2,3]. Additional constraints on Martian crustal and mantle structure have been available from meteorites, satellite probes, and direct sampling from the surface probes[4–6]. The Martian core-mantle boundary is placed at 1,520–1,840 km based on geodetic considerations[2,7]; however, the most recent seismic constraints place it at $1830 \pm 40$ km[8]. Due to the lack of more geophysical data, the Martian interior models are unsurprisingly not unique[9]. The Martian crust presents a significant spatial dichotomy[10,11] with an averaged crustal thickness of ~38–62 km, with the crust in the northern hemisphere ~25 km thinner than that in the southern hemisphere[2]. The crustal thickness beneath the InSight lander is determined to be $20 \pm 5$ km or $39 \pm 8$ from seismic observations[12], due to limited seismic constraints based on several marsquakes with a relatively high signal-to-noise ratio. Estimating the current-day crustal characteristics and its relationship with the Martian mantle is critical in geodynamical modeling and supporting or eliminating past Martian dynamics scenarios. For example, a thicker crust would imply that Mars exchanged heat with the exterior less efficiently, and the mantle convection could have stopped entirely. The InSight mission recently motivated a wide range of other studies regarding the Martian internal structure[13].

It is clear from the rock magnetization that the magnetic field once existed globally on Mars[4,14]; however, it does not exist today. Convection in planetary mantles helps the heat exchange at the core-mantle boundary and, consequently, a dynamo's operation. Therefore, one of the outstanding questions is whether the mantle is subject to convective motions at the present day or the convection ceased sometime in the past. Given the absence of a magnetic field and uncertainties on the Martian core's chemical composition[2], the Martian core could still be completely molten, completely solidified, or made up of a combination of both aggregate states[7,15]. The lack of mantle convection and thus the absence of efficient heat exchange with the core would explain the Martian core convection's termination. As the Love number inferred from Mars Global Surveyor tracking data indicates[7] the core could still be liquid due to the presence of sulfur with iron and nickel but not convecting. Recent seismological observations indeed support a liquid core[8]. Alternatively, if the mantle is mobile, there could be enough heat exchange across the core-mantle boundary for convection in the core, but the liquid core layer must be too thin to allow a dynamo operation. Because of these fundamental questions, one of the InSight mission goals is to probe the tectonic activity of Mars via measuring the magnitude and frequency of Martian seismicity, given that the prior seismometer of Viking 2[16] did not robustly observe more than a single candidate marsquake[17]. In particular, observations of continuing seismicity due to magma mobility in the mantle or lower crust would go a long way in proving that the Martian mantle is indeed mobile.

Various studies have attempted to quantify the global occurrence (although focusing on the Tharsis region) of marsquakes and annual moment release rates based on the surface distribution of faults and cooling-related contraction e.g.,[18,19]. Ref. [20] forecasted seismic activity specifically for the Cerberus Fossae graben system in the proximity of the InSight landing site. Based on the morphology of the grabens and assuming their tectonic activity, they estimated that between $1.5 \times 10^0$ and $1.9 \times 10^5$ events per year would be detectable by the InSight seismometer. Besides, the seismicity around linear features of Cerberus Fossae might be due to extensive forces caused by dyke emplacement associated with the surrounding volcanic provinces e.g.,[21,22]. Also, low-frequency marsquakes identified as subcrustal[9] were recently interpreted via a volcanic source model typical for low-viscosity, high-volume flux lavas associated with Cerberus Fossae. Thus, InSight's detection of mantle marsquakes as possible tremor episodes caused by the magma transport process would be crucial in understanding if the Martian mantle is still mobile.

From the time the seismometer SEIS[23,24], a part of the InSight apparatus, was deployed on the ground in February 2019 and March 2020, various teams successfully detected 465 marsquakes[9,17]. Initially, two types of marsquakes have been reported, high-frequency and low-frequency events[9,17]. Moreover, the third class of events was named the super high-frequency events as they are characterized by energy higher than 5 Hz and are explained as a result of thermal cracking near the InSight lander[25,26]. Of those, 424 high-frequency (HF) events have seismic energy in the frequency band around 2.4 Hz and have been characterized as sub-surficial, crustal events. In comparison, only 41 low-frequency (LF) events with dominant periods of 1–10 s originated from the Martian mantle. Although 41 marsquakes have been identified as LF events, most of them do not show clear P and S onsets like earthquakes, indicating these marsquakes are somewhat weak. Until now, only two marsquakes with clear onsets and polarities have been detected, i.e., S0173a and S0235b. The locations of the InSight lander and the two events are shown in Fig. 1.

The currently reported marsquakes are painstakingly[17] identified by the InSight Marsquake Service[27]. Due to the relatively small magnitude of marsquakes and the possible scattering or noise, there are significant uncertainties in the arrivals of identifiable phases, contributing to difficulties locating most marsquakes. These marsquakes' physical mechanisms are unknown because only a single recording station is available[9]. Furthermore,

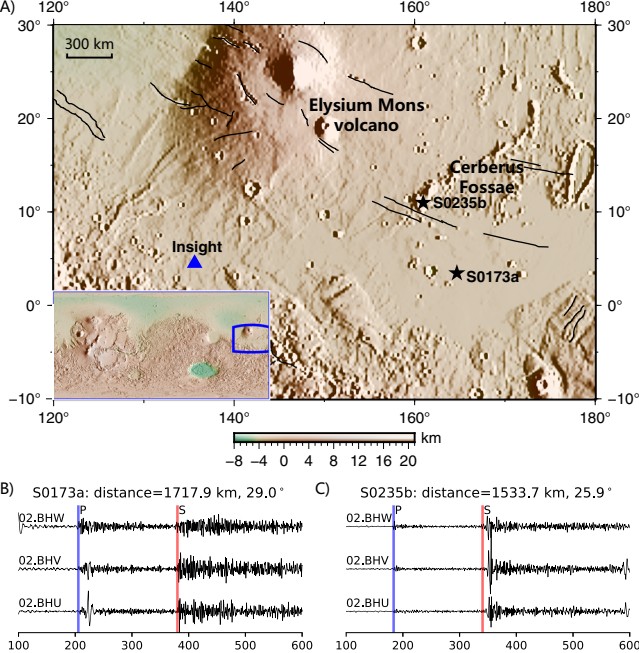

**Fig. 1 The landing site of InSight and waveforms of two quality-A marsquakes[17]. A** The landing site of InSight is marked by a blue triangle, while black stars denote the two identified marsquakes. The solid black lines demonstrate the Graben faults[19]. The lower left inset map shows the global Martian topography, and the blue rectangular illustrates the research region. **B** the bandpass-filtered three-component waveforms of the S0173a event. The blue and red vertical lines label the P-wave and S-wave arrivals. **C** same as **B** but for the S0235b event. The two-way Butterworth filter is applied to the seismograms in the band 0.1–0.8 Hz. The channel name is labeled at the top left corner.

their spatial evolution and tectonic implications have not been revealed. Consequently, the lack of recorded marsquakes prevents the deep Martian interior's illumination and makes inferences on the Martian mantle's activity more challenging.

The manual picking or automated algorithms (e.g., STA/LTA)[28] for seismic wave arrivals might prevent the detection of many small-magnitude subcrustal marsquakes hidden in ambient noise or coda of known marsquakes. Therefore, there is the need to complement the existing searches with dedicated searches for potential, smaller marsquakes buried in noisy waveforms.

In this work, we deploy two relatively unconventional search methods, recently introduced to the geophysics community: the matched filter (MF)[29–31] and the law of first digits, a.k.a. Benford's Law (BL)[32–34]. Using the existing catalog of marsquakes and these two methods, we perform a systematic search over the available continuing waveform data for marsquakes that might have indeed gone undetected. Here, we report 47 new low-frequency events that mostly resemble the waveforms of S0173a and S0235b events, assuming they are collocated. We interpret them as marsquakes. They are repetitive events, and their re-occurrence allows us to infer that Martian's subcrustal activity is substantially higher than anticipated. We describe the detection of new marsquakes and discuss possible physical mechanisms responsible for their origin.

## Results

As the InSight SEIS waveform records are characterized by a relatively low signal-to-noise ratio for the detected events, the marsquakes identification is approached carefully using state-of-the-art algorithms STA/LTA[28] and a careful verification by seismologists. The InSight Marsquake Service[27] manually checks seismic data and picks Martian events[17]. Apart from supplementing the Marsquake InSight catalog, documenting such small events is critical in the context of Mars's tectonic implications. We here start by assuming that the currently used approaches could miss small-magnitude events buried in ambient noise. We subsequently employ the MF[29–31] and BL[32–34] identification methods to detect small-magnitude events (Supplementary Note 1; Supplementary Figs. 3–50) buried in noise. As noted already, both the MF and BL approaches have been successfully utilized in seismology to detect seismic events from continuing waveforms. We elaborate on the data and provide descriptions of the methods (Methods, Supplementary Notes 2 to 4 and Figures therein).

**Repeating marsquakes**. Using the MF method (Materials and Methods, and Supplementary Notes 1 to 3), we match S-wave arrivals of the nine known low-frequency marsquakes with the continuous waveforms reported in Extended Data Fig. 4 of ref. [9] and listed in Supplementary Table 1. As a result, we detect 47 matched events in total. The high-quality confirmed marsquakes, S0173a and S0235b, yield the most repetitive events, 45 in total. Out of these, 11 are associated with S0173a and 34 with S0235b (Table 1). All other events from the MF detections are also listed in Table 1. Although we applied the BL method on all available continuous waveforms, we conclude that it gives half-successful results even for the two high-quality events due to the presence of numerous glitches[23,35] (see Supplementary Note 1.2 and Supplementary Fig. 50). We, therefore, turn our attention entirely to the undetected events search via the MF method. Since these newly detected events are not included in the manually picked marsquakes catalog[17], we further scrutinize the robustness of their detection to verify if they are indeed repetitive marsquakes (Supplementary Notes 2 and 3).

Figure 2 illustrates selected examples of the newly detected events via the MF method using the S0173a and S0235b events. Notably, two repeating events are detected immediately after the S0173a event (Fig. 2A–D). The normalized cross-correlation coefficients (CC) of all traces are higher than 0.5. We note that the threshold CC of 0.46 we adopted empirically is much more conservative than the Parkfield earthquakes threshold (CC = 0.15), as illustrated in ref. [29]. We show similar examples for the S0235b event (Fig. 2E–H), giving the threshold CC of 0.53. The newly detected events are listed in Table 1, and their waveforms are shown in Supplementary Figs. 3 to 49.

To reinforce the conclusion that the matching events are repetitive marsquakes, we also calculate the cross-correlation coefficients between the P-arrival waveforms of the template marsquakes[9,17] and the matching events (Table 1). S0173a-MF01 and S0173a-MF02 (Table 1) are examples of high CC for P-waves, with their CCs reaching 0.83 and 0.77 on the 02.BHW component. Such high CC values for both S and P waves indicate high similarities of the newly detected events with the matching marsquakes and reinforce our conclusion that they are repetitive marsquakes. However, due to the weak P-arrivals on most waveforms, we could not expect high P-wave CC values to be required for other events. We could also not entirely rule out that some of the newly detected events are not the multiple reverberates of seismic phases due to the significant uncertainties of the Martian structural models lacking more seismic constraints (More discussions in Supplementary Note 4). The complete waveforms of the newly detected events are shown in Supplementary Figs. 2 and 3 for the S0173a and S0235b matching marsquakes.

**Marsquakes in diurnal time**. Notably, only four other new marsquakes are detected by taking the additional seven B- and C-quality marsquakes as templates (see Supplementary Table 1). This might mean that the signal-to-noise ratio of template marsquakes is relatively low as they could be significantly affected by ambient noise. Previous works reported that most marsquakes occurred during quiet nights[9]. We find that 21 out of 47 repetitive marsquakes occurred during the daytime (05:00:00–17:00:00 local mean solar time, LMST) as listed in Table 1.

## Discussion

Using the MF technique and the S waveforms of the previously detected marsquakes as matching templates, we detected 47 new events. Based on a similarity of their waveforms, we conclude that they are repetitive marsquakes located at similar locations. The observation that a significant portion of these newly detected marsquakes happened during the daytime (Fig. 3 and Table 1) is crucial in deciphering their origin. Namely, a family of marsquakes with short durations was linked previously with thermal stresses[17]. Indeed, marsquakes could result from the cooling and contraction of Mars from the 3-D thermal evolution model[36]. However, these thermal-related marsquakes have a relatively high frequency, ~5–30 Hz, and occur only during a 2-hour time window around sunset due to significant temperature variations[25]. They are characterized as being near-surface, and their locations are near the InSight lander. Given the similarity with the low-frequency parent marsquakes (S0173a and S0235b) and the time of their occurrence, the newly detected marsquakes cannot be temperature-induced. Most likely, all repetitive marsquakes we observe are subcrustal, like their "parent" marsquakes[9,17].

Tide-related earthquakes and moonquakes were documented in the Earth-Moon system[37]. Phobos and Deimos, the Martian moons, also raise a tidal response to Mars[38]. Two distinct categories of moonquakes that occur in the lunar interior are

**Table 1 The newly detected events using the S0173a, S0235b, and other seven events as templates.**

| Event name | Origin (UTC) | P arrival (s) | P normalized CC BHU | BHV | BHW | S arrival (s) | S normalized CC BHU | BHV | BHW | Origin (LMST) | Term $\mathcal{A}$ (Equation 5) |
|---|---|---|---|---|---|---|---|---|---|---|---|
| S0173a-MF01 | 2019-05-23T02:19:58.011430Z | 205.46 | 0.17 | 0.29 | 0.83 | 380.15 | 0.51 | 0.60 | 0.65 | S0173 02:55:41 | −0.231 |
| S0173a-MF02 | 2019-05-23T02:20:13.211430Z | 205.46 | 0.42 | 0.63 | 0.77 | 380.15 | 0.58 | 0.65 | 0.63 | S0173 02:55:56 | −0.284 |
| S0173a-MF03 | 2019-06-15T22:55:07.561430Z | 205.46 | −0.08 | −0.11 | 0.07 | 380.15 | 0.49 | 0.38 | 0.60 | **S0196 08:11:39** | −0.109 |
| S0173a-MF04 | 2019-06-21T22:47:15.811430Z | 205.46 | 0.43 | 0.25 | −0.08 | 380.15 | 0.35 | 0.44 | 0.60 | S0202 04:12:50 | −2.306 |
| S0173a-MF05 | 2019-07-07T18:32:36.914430Z | 205.46 | −0.25 | 0.14 | −0.31 | 380.15 | 0.53 | 0.51 | 0.37 | **S0217 13:48:33** | −0.607 |
| S0173a-MF06 | 2019-07-28T16:34:07.061430Z | 205.46 | 0.12 | −0.20 | 0.15 | 380.15 | 0.43 | 0.42 | 0.54 | S0237 22:24:07 | −3.002 |
| S0173a-MF07 | 2019-11-11T01:38:37.211430Z | 205.46 | −0.23 | 0.01 | −0.06 | 380.15 | 0.30 | 0.50 | 0.66 | **S0340 11:48:36** | 0.060 |
| S0173a-MF08 | 2019-11-17T03:30:30.961430Z | 205.46 | −0.12 | 0.22 | 0.14 | 380.15 | 0.52 | 0.41 | 0.51 | **S0346 09:46:20** | −0.104 |
| S0173a-MF09 | 2020-02-12T09:16:12.411430Z | 205.46 | 0.18 | 0.09 | 0.06 | 380.15 | 0.42 | 0.46 | 0.54 | **S0431 07:30:49** | −0.156 |
| S0173a-MF10 | 2020-02-12T12:06:35.461430Z | 205.46 | −0.24 | −0.09 | 0.02 | 380.15 | 0.43 | 0.43 | 0.60 | **S0431 10:16:38** | −0.858 |
| S0173a-MF11 | 2020-02-29T20:34:44.811430Z | 205.46 | 0.47 | −0.25 | 0.08 | 380.15 | 0.43 | 0.66 | 0.37 | **S0448 07:36:13** | −0.100 |
| S0235b-MF01 | 2019-03-15T14:51:16.121357Z | 183.69 | −0.30 | 0.05 | 0.14 | 340.54 | 0.38 | 0.65 | 0.63 | **S0106 11:25:20** | −0.442 |
| S0235b-MF02 | 2019-03-17T09:49:41.921357Z | 183.69 | 0.34 | 0.16 | 0.17 | 340.54 | 0.51 | 0.71 | 0.40 | **S0108 05:14:47** | −0.596 |
| S0235b-MF03 | 2019-06-12T06:03:03.221357Z | 183.69 | −0.34 | 0.08 | −0.03 | 340.54 | 0.41 | 0.59 | 0.58 | S0192 17:42:15 | −0.622 |
| S0235b-MF04 | 2019-06-16T23:33:32.421357Z | 183.69 | 0.55 | −0.29 | −0.04 | 340.54 | 0.54 | 0.47 | 0.63 | **S0197 08:10:31** | 0.465 |
| S0235b-MF05 | 2019-06-17T14:19:53.821357Z | 183.69 | −0.01 | −0.03 | −0.06 | 340.54 | 0.46 | 0.53 | 0.55 | S0197 22:33:09 | −1.766 |
| S0235b-MF06 | 2019-06-18T07:36:43.821357Z | 183.69 | 0.37 | 0.21 | −0.20 | 340.54 | 0.25 | 0.68 | 0.38 | **S0198 15:22:15** | −0.236 |
| S0235b-MF07 | 2019-06-26T04:47:28.321357Z | 183.69 | 0.44 | −0.22 | −0.01 | 340.54 | 0.50 | 0.36 | 0.69 | **S0206 07:29:17** | −0.596 |
| S0235b-MF08 | 2019-06-28T01:44:51.321357Z | 183.69 | 0.17 | 0.22 | 0.13 | 340.54 | 0.39 | 0.59 | 0.58 | S0208 03:14:30 | −1.842 |
| S0235b-MF09 | 2019-07-17T06:51:31.421357Z | 183.69 | 0.55 | −0.25 | −0.07 | 340.54 | 0.49 | 0.41 | 0.65 | S0226 20:00:56 | −1.485 |
| S0235b-MF10 | 2019-08-02T21:16:49.771357Z | 183.69 | 0.25 | −0.02 | 0.27 | 340.54 | 0.50 | 0.66 | 0.63 | S0242 23:46:38 | −1.401 |
| S0235b-MF11 | 2019-08-03T20:15:17.171357Z | 183.69 | −0.05 | 0.40 | −0.10 | 340.54 | 0.64 | 0.48 | 0.47 | S0243 22:08:12 | −2.168 |
| S0235b-MF12 | 2019-10-22T01:49:36.421357Z | 183.69 | 0.34 | −0.12 | −0.03 | 340.54 | 0.54 | 0.52 | 0.67 | S0321 00:49:51 | −1.657 |
| S0235b-MF13 | 2019-10-31T06:22:43.271357Z | 183.69 | −0.03 | −0.10 | 0.07 | 340.54 | 0.44 | 0.59 | 0.62 | S0329 23:28:54 | −2.034 |
| S0235b-MF14 | 2019-11-01T10:25:15.471357Z | 183.69 | −0.24 | −0.12 | 0.25 | 340.54 | 0.55 | 0.58 | 0.57 | S0331 02:46:25 | −1.185 |
| S0235b-MF15 | 2019-11-01T16:22:17.771357Z | 183.69 | 0.27 | −0.04 | −0.02 | 340.54 | 0.37 | 0.56 | 0.71 | **S0331 08:33:54** | 0.018 |
| S0235b-MF16 | 2019-11-05T23:29:58.721357Z | 183.69 | −0.07 | 0.01 | 0.33 | 340.54 | 0.60 | 0.55 | 0.37 | **S0335 12:56:02** | 0.658 |
| S0235b-MF17 | 2019-11-07T10:01:31.771357Z | 183.69 | 0.27 | 0.09 | 0.02 | 340.54 | 0.51 | 0.54 | 0.52 | S0336 22:32:10 | −2.380 |
| S0235b-MF18 | 2019-11-08T10:15:11.821357Z | 183.69 | −0.36 | −0.20 | −0.33 | 340.54 | 0.47 | 0.55 | 0.55 | S0337 22:06:56 | −2.096 |
| S0235b-MF19 | 2019-11-16T15:52:13.421357Z | 183.69 | 0.02 | 0.44 | 0.12 | 340.54 | 0.38 | 0.57 | 0.51 | S0345 22:26:43 | −2.365 |
| S0235b-MF20 | 2019-11-17T21:39:35.671357Z | 183.69 | 0.08 | 0.18 | −0.02 | 340.54 | 0.40 | 0.58 | 0.65 | S0347 03:26:16 | −1.081 |
| S0235b-MF21 | 2019-11-20T10:14:49.471357Z | 183.69 | −0.23 | −0.05 | −0.05 | 340.54 | 0.26 | 0.74 | 0.65 | **S0349 14:24:14** | −0.070 |
| S0235b-MF22 | 2019-11-27T20:44:20.121357Z | 183.69 | 0.18 | −0.24 | 0.09 | 340.54 | 0.47 | 0.51 | 0.67 | S0356 20:07:12 | −2.025 |
| S0235b-MF23 | 2019-12-01T03:11:35.621357Z | 183.69 | 0.14 | 0.27 | 0.18 | 340.54 | 0.53 | 0.54 | 0.62 | S0360 00:28:31 | −1.380 |
| S0235b-MF24 | 2019-12-03T23:14.671357Z | 183.69 | −0.44 | −0.04 | 0.00 | 340.54 | 0.48 | 0.52 | 0.61 | S0361 23:22:48 | −2.128 |
| S0235b-MF25 | 2019-12-10T22:36:41.521357Z | 183.69 | −0.43 | −0.38 | 0.04 | 340.54 | 0.54 | 0.57 | 0.55 | **S0369 13:35:41** | −1.243 |
| S0235b-MF26 | 2019-12-18T04:16:11.271357Z | 183.69 | −0.22 | −0.04 | 0.20 | 340.54 | 0.49 | 0.46 | 0.74 | **S0376 14:36:24** | −0.477 |
| S0235b-MF27 | 2019-12-25T22:57:11.021357Z | 183.69 | −0.38 | 0.30 | −0.06 | 340.54 | 0.41 | 0.64 | 0.60 | S0384 04:17:42 | 0.160 |
| S0235b-MF28 | 2020-01-17T12:14:16.871357Z | 183.69 | −0.29 | 0.19 | 0.27 | 340.54 | 0.50 | 0.47 | 0.60 | S0406 03:05:51 | −2.095 |
| S0235b-MF29 | 2020-01-18T23:24:15.121357Z | 183.69 | 0.28 | 0.13 | 0.00 | 340.54 | 0.50 | 0.45 | 0.52 | **S0407 13:19:22** | 0.143 |
| S0235b-MF30 | 2020-01-23T13:53:54.921357Z | 183.69 | −0.18 | −0.25 | 0.33 | 340.54 | 0.50 | 0.49 | 0.63 | S0412 00:51:39 | −1.900 |
| S0235b-MF31 | 2020-02-07T12:20:02.421357Z | 183.69 | −0.08 | 0.28 | 0.03 | 340.54 | 0.43 | 0.68 | 0.50 | **S0426 13:42:22** | −1.694 |
| S0235b-MF32 | 2020-02-14T21:01:17.171357Z | 183.69 | 0.52 | −0.36 | 0.07 | 340.54 | 0.45 | 0.51 | 0.63 | **S0440 06:40:43** | −0.435 |
| S0235b-MF33 | 2020-02-23T23:01:05.471357Z | 183.69 | 0.26 | 0.01 | −0.08 | 340.54 | 0.57 | 0.42 | 0.80 | **S0442 13:49:49** | −0.490 |
| S0235b-MF34 | 2020-03-19T20:47:29.130357Z | 183.69 | 0.36 | −0.27 | −0.05 | 340.54 | 0.55 | 0.38 | 0.78 | S0466 19:36:35 | −2.773 |
| S0183a-MF01 | 2019-03-18T01:51:04.613000Z | 326.80 | −0.45 | 0.52 | 0.25 | 590.00 | 0.62 | 0.52 | 0.54 | S0108 20:50:26 | −0.702 |
| S0325a-MF01 | 2019-03-18T04:42:06.514938Z | 287.98 | −0.16 | 0.01 | 0.36 | 525.34 | 0.61 | 0.51 | 0.55 | S0108 23:36:53 | −0.769 |

The Origin (UTC times) are listed. The P and S arrivals are relative to the origin. The normalized cross-correlation coefficients of each channel are also given for both P and S arrivals. The last column shows the estimated event origin in local mean solar time (LMST). The events that occurred during the Martian, noisier daylight, are bolded. The daylight approximately ranges from 05:00:00 to 17:00:00 LMST following ref. 9 shown in Fig. 3. The term $\mathcal{A}$ containing amplitude ratios is calculated according to Eq. (5). The events that occurred during the diurnal time, are bolded.

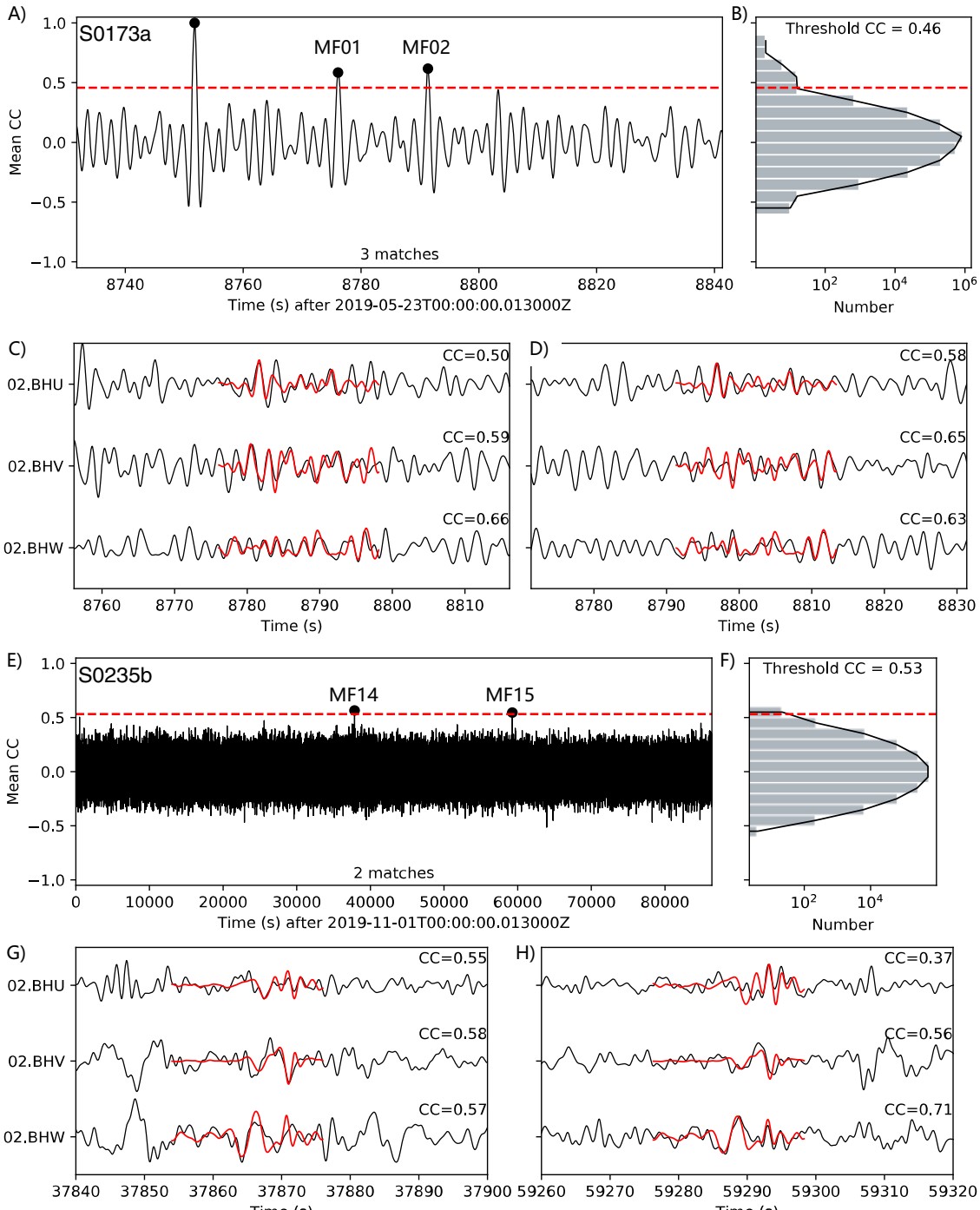

**Fig. 2 Two examples of the newly detected marsquakes. A** The averaged cross-correlation function for the S0173a event is used as a matched filter. The dots illustrate the timings of positive detections of marsquakes above the threshold (red dashed line). The dot at CC = 1.0 shows the self-detection of the S0173a event. **B** The histogram of the averaged cross-correlation functions with the same vertical axis as **A**. **C**, **D** The template and continuous-waveforms match for S0173a-MF1 and S0173a-MF2, also listed in Table 1. Red lines are the template event waveforms. The trace names are labeled on the left. The cross-coefficients of each component are listed on the right. **E**–**H** same as **A**–**D** but for the S0235b template event and the matches S0235b-MF14 and MF15; see Table 1.

identified as deep and shallow moonquakes. A large number of deep moonquakes (more than 12,000) located in the lower lunar mantle (500-1000 km) were considered to be from tidal stress variations caused mainly by Earth[39]. In contrast to a large number of deep moonquakes, only 28 shallow moonquakes are observed in the lunar upper mantle with a depth range of 50-200 km and magnitudes of ~5.0, which could be a result of young thrust faults[40]. Most marsquake magnitudes are comparable with

deep moonquakes (M = ~3.0 or smaller)[39], but marsquakes are located in the upper rather than the lower Martian mantle. The lack of tidal periodicity (Fig. 3) also rules out the possibility that the tidal modulation – found feasible for shallow marsquakes in water-filled confined aquifers regime[37] – affected LF marsquakes (Fig. 3).

Ref. [17] reported 465 marsquakes from February 2019 to March 2020, while ref. [9] identified 174 marsquakes from February 2019

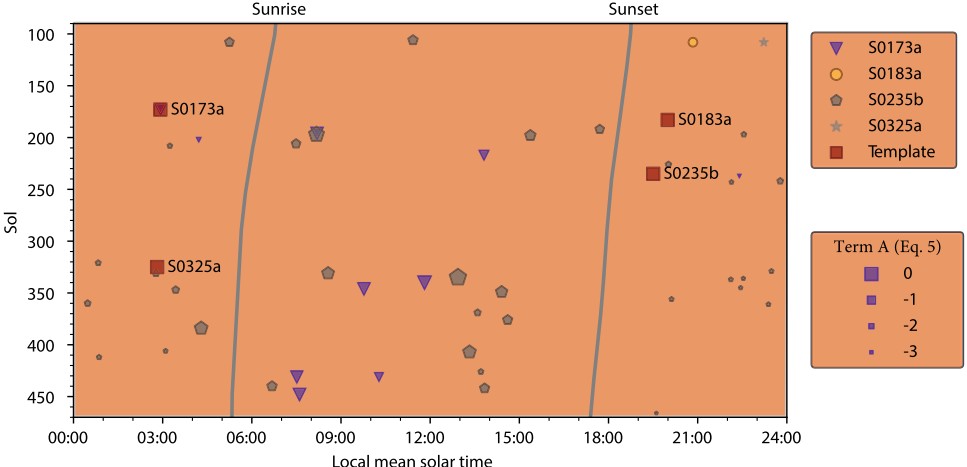

**Fig. 3 The temporal distribution and the amplitude relationship of all newly detected MF events relative to the template events.** For the exact times, refer to Table 1. The time interval ranges from Sol 90 to Sol 470. The thick lines show the sunrise and sunset time on Mars. The colored symbols denote MF events with parent (template) marsquakes listed on the right. The symbol size denotes the estimated term $\mathcal{A}$ of the marsquakes defined in Eq. (5).

to September 2020. The two A-quality marsquakes, S0173a and S0235b, that yielded 11 and 34 repetitive events in our study are traced to the Cerberus Fossae region. According to recent studies of the Cerberus Fossae, these events could be related to tectonic stress[21,41], thermo-elastic cooling[19,20,36], fluid migration[9] or active magma flow[42]. Indeed, Cerberus Fossae is geologically younger than 20 Ma[20], extending for ~1,200 km across the Martian surface[6], and faults showing ~500 m vertical offset in the topographic studies[6,43]. The long-term tectonic activities of Cerberus Fossae are supported by the observations of boulder trail populations with lengths up to tens of kilometers from Mars Orbiter Laser Altimeter (MOLA) images of Mars[5] or the High-Resolution Imaging Science Experiment (HiRISE) images[6]. The mobilization of these boulders (~150–240 km in length) was considered a result of ground shaking from paleo-marsquakes with a moment magnitude of ~7.5[5,6]. The observations of repetitive marsquakes in this region add to the notion that it is indeed geologically active. However, the substantial depth of these marsquakes[9] and their high repetitive nature make it less plausible that they are associated with tectonic activities on normal faults.

The newly detected marsquakes are repetitive events of the observed template events or the events that might have occurred before the SEIS installation. The S0173a and S0235b events have the largest number of repetitive marsquakes, while none of the repetitive marsquakes are matched for the other four events: S0105a, S0133a, S0154a, and S0189a, with a threshold of seven times of median absolute deviation (MAD). Our amplitude ratio estimates (see Methods and Eqs. (4) and (5)) indicate that, out of 34 newly detected events, S0235b is followed by 30 events with amplitude ratios smaller than 1.0. Similarly, S0173a is followed by 10 events with amplitude ratios smaller than 1.0. Still, the relatively small numbers of newly detected events preclude more in-depth statistics. However, our findings suggest that Martian seismicity is dominated by prolonged activity in only specific regions.

The S0235b event has more repetitive marsquakes than the S0173a. This may indicate the spatial heterogeneities of tectonic stress along with the Cerberus Fossae fault system, supported by similar observations on Earth[44,45]. Its first aftershock, S0235c, follows in about 35 min[9]. Still, the MF technique's cross-correlation does not show high waveform similarities between S0235b and S0235c. That might mean that the S0235c event is located slightly farther away from S0235b but could also indicate

the Martian upper mantle's strong lateral heterogeneities. The mere existence of aftershocks (with or without high waveform similarities) suggests continuous or intermittent stress accumulation and release.

On Earth, about 90% of earthquakes are distributed along plate boundaries mainly due to tectonic activities such as plate motion, while ~10% of events occur within the interior of lithospheric plates[46]. In contrast to the Earth, which hosts active plate tectonics, Mars is a single-plate planet with stagnant mantle convection[14]. Since Mars has a single plate, it is plausible to compare marsquakes with intraplate earthquakes. The possible mechanisms of intraplate earthquakes apart from slip on faults include volcanism and underground fluid slumping[47].

Although it is difficult to locate most marsquakes due to noisy signals recorded by a single station, they could be caused by volcanic activity given the observations of dark pits and haloes from Mars Orbiter Camera images of the Cerberus Fossae region that suggest ongoing volcanism[6]. Volcanic earthquake swarms on Earth are commonly observed, resulting in volcanic eruptions. An eruption did not occur on Mars during the InSight project; therefore, the volcanic activity could be due to the prevalent intrusive magmatism. Ref. [48] study ended concluding that a combination of the thick crust, spasmodic magma replenishment, crustal heating, and limited quantities of magmatic water and carbon dioxide are the main reasons for the suppressed eruptibility. In fact, in both extension and compression regions, magma could be prevented from reaching the surface. In the regions of high extensional strain, most magma could be emplaced in the crust due to the space created by the strain, and in the regions of high stress, a thickened crust will reduce the extrusion. In Ref. [42] the Cerberus Fossae lava properties were compared with the Earth-like volcanoes. Although the recent Martian lava flows are substantially different from their Earth counterparts, the conclusion was that a low-viscous and high-volume flux magma activity as a cause of low-frequency marsquakes could not be ruled out.

Apart from quakes caused by magma migration, other types of volcanic events that have recently been discovered are deep, low-frequency earthquakes that occur at quiescent volcanoes due to the repeated pressurization of volatiles in subcrustal magma[49]. Apart from ascending magma, even stalled subcrustal intrusions continue degassing through the process of secondary boiling. Recent documented earthquakes beneath Mauna Kea share at least one common characteristic with the Cerberus Fossae region

marsquakes detected here – they are rich in low-frequency content. This has been found for many other dormant Earth volcanoes in the lower crust and upper mantle[50], though this does not necessarily indicate the low-frequency marsquakes uniquely result from volcanic activities.

The number of marsquakes recorded during both the day and night in Fig. 3 after the InSight landing suggests that Mars' interior is in motion and that the Martian seismicity is continuous and long-term. The frequency and magnitudes of the newly observed marsquakes indicate that the Martian mantle might be more dynamic than anticipated based on initial observations, which is also strongly supported by identifying a low-velocity zone in the upper mantle from seismic observations[9,51]. Another piece of supporting information comes from the geochemical studies of zircons of northwestern Africa's 7034/7533 meteorites[52]. The chondritic-like Hf-isotope compositions of these zircons indicate a primitive and convecting mantle[52]. Further, the U-Pb ages of zircon are classified into two groups, i.e., ~4.3 Ga and 1548 Ma to 299.5 Ma. The younger produced zircon is thought to be from long-lasting magmatic activities in the Tharsis and Elysium provinces, two deep-seated mantle plumes[52]. If the meteorites are from Elysium Mons, the marsquakes in the Cerberus Fossae are ~600 km to 1000 km away (see Fig. 1).

Although we cannot rule out the tectonic causes, the repetitive nature of marsquakes has its equivalence in repetitive tremors in magma transfer systems on Earth – our preferred interpretation of the newly observed events' origin (Fig. 4). Last but not least, the forensic techniques applied here can detect marsquakes in different seasons and parts of sols, which plays a crucial role in planning future planetary missions.

## Methods

**Data**. NASA's Interior Exploration using Seismic Investigations, Geodesy and Heat Transport (InSight) Lander was launched on May 5, 2018, from Vandenberg Air Force Base, California. After about half a year, it landed on November 26, 2018, at Elysium Planitia, near Mars' equator on the western hemisphere of Mars (see the inset of Fig. 1). The Martian seismometer, known as the Seismic Experiment for Interior Structure (SEIS), was sitting on the lander's deck until December 19, 2019. The SEIS was then placed on the Mars surface by a robotic arm, 1.636 meters away from the lander. After adjustments to the seismometer in the next several weeks, a Wind and Thermal Shield was placed over SEIS to shield it from ambient wind noise on Feb 4, 2019.

The InSight seismic data was accessed via InSight Mars SEIS Data Service[53]. The InSight seismic station contains three short-period (SP) and three very broadband (VBB) sensors to record separately high- and low-frequency events.

The high-frequency data are more affected by ambient noise (e.g., wind and local noises), making the implementation of the marsquake-search methods featured in this manuscript more challenging. Furthermore, the high-frequency events that are readily recorded have been shown to reflect the relatively shallow near-surface structures around the InSight lander[26]. We, therefore, concentrate on the low-frequency seismic events penetrating through the deep crust and mantle. We use the very-broadband seismic data (02.BHU, 02.BHV, 02.BHW) from

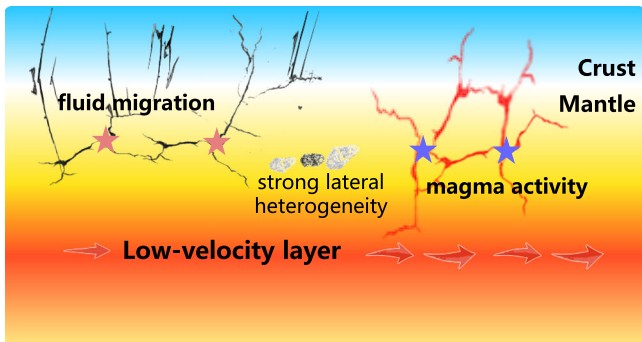

**Fig. 4 Illustration of Martian subsurface and processes that might be responsible for seismological observations.** The vertical and horizontal axes are not to scale.

12 February 2019 to 31 March 2020. The sampling rate of the three-component data is 20 Hz, producing the Nyquist frequency of 10 Hz. Those low-frequency events generally have a small magnitude between Mw 3.0 and 4.0[9]. Two A-quality marsquakes (S0173a, S0235b) are illustrated in Fig. 1B, C.

The initially retrieved data was segmented, the segments having an irregular length. We first remove the linear trend and mean value of the traces. The segmented data is then merged and split into 1-day-long time intervals in accordance with the Earth time.

**Benford's Law**. Since the signal-to-noise ratio or recorded marsquakes is generally low[9], we apply Benford's Law (BL) and matched-filter (MF) methods to detect new marsquakes. Both approaches have been successfully used to detect earthquakes. Sambridge et al.[32] tested BL on various datasets from a broad spectrum of physical sciences, including seismology, and found that earthquake recordings follow BL due to their increased dynamic range, i.e., the ratio between the smallest and the largest recorded value of ground motion. Seismic noise often has diminished dynamic range and does not follow BL. This difference serves as the basis for earthquake detection based on only the first digits of the seismogram time series. For example, the authors of[32] detected a previously unobserved, small local Canberra event preceding the Sumatra-Andaman earthquake. This was likely the first-ever earthquake detected using BL.

The BL prediction is based on a probability distribution of the first digit in a dataset. According to BL, the first digit's frequency distribution (i.e., the first non-zero number from the left) follows a logarithmic function, namely $P(D) = \log_b\left(1 + \frac{1}{D}\right)$, where D is the first digit, and b is the base of the data. In other words, lower first digits (1, 2, …) occur more frequently than the higher ones (… 8, 9). The goodness of fit measure to BL predictions is given as

$$\varnothing = \left[ 1 - \left( \sum_{D=1}^{9} \frac{(n_D - nP_D)^2}{nP_D} \right)^{1/2} \right] \times 100, \qquad (1)$$

where $n_D$ is the number of observed data with first digit D, and $n$ is the total number of data.

Here, we follow the procedures of ref. [32].

We run the BL method on the 1-day-long broadband 02.BH components with only basic preprocessing (without filtering) as described in the previous subsection. The time window length is 20 s (400 samples in total) with a sliding step of 1 s. Glitches contaminate the data due to the harsh conditions at the Martian surface, strong wind, and temperature variations. Most glitches last ~20 s on the very-broadband UVW components[35]. The ground motion time series during the two analyzed marsquakes (S0173a, S0235b) follows a BL distribution as given in Supplementary Note 1.2 and Supplementary Fig. 50. However, our analysis reveals that the BL method is susceptible to large-amplitude glitches in the Martian waveform data. Therefore, we use it to assist the detection of the aftershocks or repeated noise-masked marsquakes.

**Matched-filter method**. The matched-filter (MF) method has been successfully applied to detect both aftershocks of large-magnitude earthquakes[29], weak non-volcanic tremors, and low-frequency earthquakes[30]. The method has also been used in conjunction with the deep lunar events[54] and super high-frequency (~5–30 Hz) thermal events of InSight SEIS data[25]. The success of MF is predicated on the assumption that the high similarities between the two ground-motion waveforms may exist if the two marsquakes are located at similar locations, i.e., their spatial separation is smaller than the typical wavelength at which they are observed.

**Data processing**. We employ the MF method to detect the marsquakes with relatively small magnitudes that are "buried" in ambient noise, thus challenging to identify by the more traditional earthquake methods. The MF method consists of first taking the known event as a template. This step is then followed by matching the template with the continuous data searching for repetitive events or aftershocks with high waveform similarities via normalized cross-correlation.

The raw continuous broadband data are organized in non-standard orientations, i.e., U, V, and W. Since the template and continuous Martian field-data are processed using precisely the same procedures before cross-correlation, the search for the repetitive marsquakes is administered on the U, V, and W components without the need to remove instrumental response and rotate to Z, N, and E components.

Since the buried marsquakes are relatively weak, we must apply a bandpass filter on the Martian data to enhance signals further. The two high-quality marsquakes so far demonstrated unambiguous seismic arrivals and their polarities in the frequency band of 0.1–0.8 Hz (Fig. 1; referred to as "quality A" in Clinton, et al.[17]). We, therefore, employ a two-way fourth-order, 0.1–0.8 Hz, Butterworth bandpass filter to the template and continuous waveforms (before cutting them in smaller time intervals).

We then consider the template events or known marsquakes, which were manually detected. Nine marsquakes (listed in Supplementary Table 1) display relatively strong energy and illustrate both P and S arrivals from broadband components. They have been considered low-frequency events LFEs,[9] We take the

S arrival of these events to match the continuous data from 12 February 2019 to 31 March 2020.

One of the reasons for taking the S phases instead of P phases as template waveforms is that S phases have a significant impulse and thus have a high signal-to-noise ratio. The other fundamental reason is that the S arrivals have a much broader frequency band than the P arrivals, at least for S0173a[17]. The successful identification of repeated marsquakes mainly depends on the frequency band and the time matching window's length[55]. Empirically, we cut the time window in the intervals starting from 2 seconds before and ending at 20 s after the picked S-wave arrival time (see Supplementary Table 1). As marsquakes' frequency is lower than in their Earth's counterparts, the time window's length is empirically determined to be much longer than what is typically used for earthquakes (e.g., 4 s in ref. [29]). This is done to help improve the reliability of the newly detected marsquakes.

We then match the three-component template with the continuous broadband waveforms and calculate the cross-correlation coefficients. In the next step, we sum the coefficients of three-component channels and compute the median absolute deviation (MAD) of the mean cross-correlation coefficients to evaluate the template and continuous data's similarities. The MAD is calculated by

$$MAD = median\left(\left|\sum_{i=1}^{N} CC_i\right|\right), \quad (2)$$

where CC is the normalized cross-correlation function between a given trace and a given template. The normalized CC calculation is given as[56]:

$$CC[v(t_v), w(t_w)]_{N,\Delta_t} = \frac{\langle v(t_v), w(t_w)\rangle_{N,\Delta_t}}{\sqrt{\langle v(t_v), v(t_v)\rangle_{N,\Delta_t}\langle w(t_w), w(t_w)\rangle_{N,\Delta_t}}}, \quad (3)$$

where $v(t_v)$ and $w(t_w)$ are the discrete time-series, $\langle \rangle$ denotes the inner product, $N, \Delta_t$ are the sampling number and interval. The normalized CC will be in the range of $[-1, 1]$.

Given that only a single seismic station is deployed, the total number of traces N is 3, i.e., 02.BHU, 02.BHV, and 02.BHW for each template. Empirically, we consider "a detection" successful if the threshold exceeds MAD by seven times.

**Magnitude of MF detections**. The non-traditional magnitude of a detected event $M_{detected}$ is determined as[29,57]

$$M_{detected} = M_{template} + \mathscr{A}, \quad (4)$$

where $M_{template}$ is the reported magnitude of the template marsquakes. The term $\mathscr{A}$ is given as

$$\mathscr{A} = 2\log_{10}\left(\frac{\mathcal{A}_{MD}}{\mathcal{A}_{MT}}\right). \quad (5)$$

$\mathcal{A}_{MD}$ and $\mathcal{A}_{MT}$ represent the median values of the maximum amplitude of the detected and template events, respectively[29]. The median is defined as the middle order value of the maximum amplitudes measured on the three seismogram components, which helps to avoid small disturbances occurring on a single component[35].

The magnitude of the MF detection in Eq. (4) is quite different from the traditional definition. For the template events, the term $\mathscr{A}$ equals zero. More information can be found in the Supplementary Note 1 of the Supplementary material. We perform various robustness tests of the MF method, including parameter choices in Supplementary Notes 2.

## Data availability
All InSight SEIS data used in this paper are available in IRIS at https://www.iris.edu/hq/ or via InSight Mars SEIS Data Service at https://doi.org/10.18715/SEIS.INSIGHT.XB_2016. The processed datasets are available from the authors on reasonable request.

## Code availability
The codes are available from Supplementary Software.

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

## Acknowledgements

We acknowledge NASA, CNES, their partner agencies, and Institutions (UKSA, SSO, DLR, JPL, IPGP-CNRS, ETHZ, IC, MPS-MPG), the flight operations team at JPL, SISMOC, MSDS, IRIS-DMC, and PDS for providing SEED SEIS data. This research is supported by the National Natural Science Foundation of China (Grant no. 42022026, 41720104006 and 41774060), and the Youth Innovation Promotion Association CAS. The support from the Key Research Program of the Institute of Geology & Geophysics, CAS (IGGCAS-201904) is also acknowledged. H.T.'s time on this project is supported through a combination of grants administrated by the Australian National University. The Supercomputing Laboratory of Institute of Geology and Geophysics, Chinese Academy of Sciences is thanked for the computational resources.

## Author contributions

W.S. and H.T. designed the project, aims, and methods. W.S. carried out the data analysis. Both authors contributed to the discussion of results and paper writing.

## Competing interests

The authors declare no competing interests.
