## [Peer Review File · Nature Communications]

Repetitive marsquakes in Martian upper mantleREVIEWER COMMENTS

Reviewer #1 (Remarks to the Author):

Review summary:

This is an interesting manuscript proposing that marsquakes occurs in repeating way . InSight/SEIS data are furthermore not easy to understand, due to the extreme environment and the very low amplitude of both the seismic noise and observed marsquake.

However, the main results cannot be understood, based on simple relation between the magnitude of the proposed events, their cross-correlation coefficients with the parent's events and their expected signal to noise ratio.

I unfortunately suspect that the MF method is possibly corrupted by noise background, including glitches, and that the MF method is therefore more controlled by the similarities in the noise than by similarities in the quakes. This noise is indeed not a stochastic noise. It is strongly thermally controlled and is repeating, including for glitches, regularly.

The authors must demonstrate, for example by using deglitched data, that the same results are retrieved. The authors must also ensure that the robustness of the method with respect to noise is found in figure summarizing for example the evolution of cross-correlation coefficients with the expected signal to noise of the repeating quakes.

My recommendation is therefore to request the authors to re-submit their work with a much stronger discussion on the effects of glitches structure in their analysis, which might need to use deglitched data. In addition, the Table 1 must be associated by summary figure, which shows that a priori relations between cross-correlation coefficients, event magnitude and local time noise, are coherent with what we can expect from waveform matching of low signal to noise events. At this time, this is missing, and the reader just cannot understand the results.

Because I was therefore not convinced by the main results summarized in Table 1, I did not went in depth in the section describing the consequences in term of Mars tectonics.

Minor comments:

Line 15: The wording "Martian seismic data recorded by InSight are rather noisy" seems surprising. Mars noise is very significantly lower than the Earth micro-seismic noise, so with Earth standard, the Martian seismic data are very low noise. It seems to me that the issue is more that Marsquake have low magnitude and low amplitude, leading to low SNR.

Line 28: k2 has also provided estimate of the internal structure. See for example Smrekar et al., Pre-mission InSights on the Interior of Mars, Space Sci Rev, 215, 3, doi: 10.1007/s11214-018-0563-9, 2019

InSight/SEIS presentation: It is very much surprising that this paper is not citing any mission or instrument papers when it is introducing the InSight mission (line 28) nor the InSight seismometer (line 69). I suggest the authors to cite:

For the InSight mission:

Banerdt et al, Initial results from the InSight mission on Mars, Nature Geoscience, 13, 183-189, doi: 10.1038/s41561-020-0544-y, 2020

For the SEIS experiment:

Lognonné, P., Banerdt, W.B., Giardini, D. et al., SEIS: Insight's Seismic Experiment for Internal Structure of Mars, Space Sci Rev, 215, 12, doi : 10.1007/s11214-018-0574-6 , 2019.

Lognonné et al., Constraints on the shallow elastic and anelastic structure of Mars from InSight seismic data, Nature geoscience, 13, 213-220, doi: 10.1038/s41561-020-0536-y (2020).

Line 53: For the Viking 2 mission: citing a paper which in fact recite the Viking 2 mission paper is not the best way to present facts. The authors shall consider citing the Viking 2 mission paper instead D.L. Anderson, W.F. Miller, G.V. Latham, Y. Nakamura, M.N. Toksoz, A.M. Dainty, F.K. Duennebieer, A.R. Lazarewicz, R.L. Kowach, T.C. Knight, Seismology on Mars. J. Geophys. Res. 82, 4524–4546 (1977a) <https://doi.org/10.1029/JS082i028p04524>

Line 104: In addition, the SEIS data are not cited when used. See guidelines in the SEIS data depositories and please cite correctly these data as: InSight Mars SEIS Data Service. (2019). SEIS raw data, Insight Mission. IPGP, JPL, CNES, ETHZ, ICL, MPS, ISAE-Supaero, LPG, MFSC. https://doi.org/10.18715/SEIS.INSIGHT.XB_2016

Line 106: Note that the work is made by the MarsQuake Service team, which is hosted by ETH Zurich but is not the ETH Zurich team. See for example the MQS composition in <https://doi.org/10.12686/a11>

Line 108: The same is true for the InSight Mars Quake catalogue, which must also be cited when used, with citation as: InSight Marsquake Service (2021). Mars Seismic Catalogue, InSight Mission; V6 2021-04-01. ETHZ, IPGP, JPL, ICL, MPS, Univ. Bristol. <https://doi.org/10.12686/a11>

Line 116: Fig S1 , first figure shown, is poorly introducing the data. No unit are shown. I suspect that these are DU. At least some value of the DU is needed or alternatively, the data shall be plotted in physical unit, as it is mandatory to provide to the reader some idea of the amplitude of both the noise and signal, which is completely absent from this paper.

Line 116-126: The authors use a large fraction of their main text to say why the BL approach is not working due to glitches. But they do not indicate in the main text why the MF method is working. I therefore recommend them to do the opposite. One line to say that BL is not working and 9 lines to summarize why MF is working.

Line 125: The two papers focused on InSight/SEIS glitches are missing from the reference and shall provide more background on the origin of these glitches, mostly associated to the extremely large temperature variations of on Mars (almost 100C for the tether linking SEIS to the lander for example). See Lognonné et al., 2020 cited above and Scholz et al (2020), Detection, Analysis, and Removal of Glitches From InSight's Seismic Data From Mars, Earth and Space Science, 7, e2020EA001317, doi: 10.1029/2020EA001317

In addition, the authors shall consider to perform a deglitching of the data to ensure that glitches patterns are not corrupting their results.

Line 129: while seismologists will have a look on the supplementary material, most of the reader will first read the table and will ask themselves how these results can be valid. The very tricky point are the magnitude. These magnitudes range from a minimum of 0.7 to magnitude of 4, so therefore more than the magnitude of the detected 173a and 235b. I recall that the magnitude of the detected 173a-235 b was about 3.5.

This is leading to several questions:

- What is the magnitude definition used by the authors?
- If this is a classical body wave magnitude definition, this suggest that the lowest magnitude events will have an amplitude $10^{2.8}$ lower than the 173a-235b, which means 630. The SNR of 173a and 235b are respectively about 100 and 250. Even if the background noise depends significantly on the local time, it seems not obvious that matching can provide good results for SNR much less than one, which will be what we expect for magnitude less than 1.5
- How to explain that the event S0235b-MF04, which is proposed as magnitude 4 and with correlation coefficients of 0.54 0.47 0.63, has correlation coefficients finally comparable to event S0235b-MF34 , with 0.55 0.38 0.78 for the correlation coefficients but with a magnitude of 0.7

It seems to me that these observations on the results proposed are not internally compatible. I

imagine that the correlation coefficients are decreasing significantly with the magnitude and/or with the a priori signal to noise which will be roughly proportional to $10m/n(LMST)$, where n is noise as $LMST$. This seems not the case and therefore generate a lot of interrogation on the main message of the paper. For that reason, I believe that this analysis is not yet mature for publication.

Reviewer #2 (Remarks to the Author):

This paper is describing a method to find Martian quakes occurring at the same place than other martian quakes already detected by InSight mission.

This method is applied to the continuous recordings of SEIS instruments, and the authors claim to have found a large number of quakes identical to the two largest seismic events recorded by SEIS instrument.

Before and after the presentation of the main results, the author elaborate what are the implications of such repeated signals, interpreted as Mars Quakes, for Mars tectonics, volcanism and internal dynamics.

I will not review here the implications of these results for Mars internal dynamics.

The main reason being that the strong weak point of the paper is the reliability of these results (repeating signals on SEIS seismometer).

First, the method presented here contains some missing information that do not allow to reproduce the results, and some arbitrary parameter choices that strongly influence the results. The threshold used to validate a detection is set arbitrarily to 7 times the "MAD". Why seven? how do justify it? Probably better to use a statistical argument based on distribution of MAD. This threshold appears to vary significantly on figures provided as supplementary material. Why? Is it due to the varying size of the window used to compute the MAD? please provide the size of this window to ensure that the results can be reproduced.

Second, the robustness of the detection method is questionable for the following reasons. No synthetic examples are provided. Robustness of the detection is not tested for different window sizes and/or different S wave arrival times. Most the detections based on S wave are providing very low correlation coefficients for P waveform comparison. Can you show how the P amplitude (by scaling the reference waveforms) would compare to signals assumed as P? Compute an SNR? If the method is working, you would expect that the stack of the waveforms would converge to the waveform of the reference event, like it was done for the deep Moon quakes. This simple test is not even tried by the authors.

Third, there is some wrong interpretation of the input data and the results of the method. The frequency range used here (0.1-0.8 Hz) is going to too low frequencies, in a region (0.1-0.2 Hz) presenting much more noise than above 0.3 Hz. This noise is driven by a combination of Pressure, Wind effects and glitches that can repeat due to atmospheric forcing and temperature periodicity of glitches. Due to the S wave pick for S0235b being too early by more than 5s, the long period noise before the S arrival, but in the S window used, is dominating the correlation coefficient estimates. This is the reason why so many quakes are detected for that particular event (that particular S waveform). In addition, using the inclined U,V,W components, and not ZNE, give more weight to the tilt noise sources expressing mainly along the horizontal components below 0.3 Hz.

Concerning the results, events S0173a-MF01 and S0173a-MF02 correspond to the SS and SSS multiples of the S wave of S0173a. S-P, SS-S (S0173a-MF01 minus S0173a) and SSS-S (S0173a-MF02 minus S0173a) differential of this event can be consistently explained by ray tracing in Mars internal structure models.

Finally, the authors observe moment magnitudes larger during day time than during night time. This observation can be easily explained if you assume that the method is fitting noise, because the amplitude of the noise is much larger during the day.

Even if the method and results are not convincing, I would encourage the authors to pursue in their studies using match filtering, targeting on the resolution of the above mentioned issues.

I provide a manuscript with some comments.

Major revision.

Additional comments:

- the InSight catalog is produced by Mars Quake Service (in which at least 5 different organisations are involved) not by ETHZ team.

Reviewer #3 (Remarks to the Author):

NCOMMS-21-18238 Review: Repetitive marsquakes in the martian upper mantle

Summary:

The authors apply two signal processing techniques to identify repetitive marsquakes in InSight data. Using Benford's Law-based statistics, and matched filter templates, they report detection of 47 new seismic events from the Cerberus Fossae region of Mars. From this, they conclude that the martian mantle is still mobile and seismically highly active.

Overall comments:

Benford's Law and matched filter banks are not, as far as I know, standard techniques currently used to identify marsquakes in InSight data. Therefore, if they can be shown to robustly identify events and exclude false positives, they have significant potential in terms of application to martian seismic data.

However, at present it is unclear whether the results presented herein are robust and reliable. Some of the conclusions I find extremely surprising, especially with regard to the lack of diurnal patterns in the detected marsquakes. Whilst this does not mean that the techniques are incorrectly applied or invalid, it does mean that they are not thoroughly presented enough to be convincing.

In terms of structure, the manuscript is reasonably logical and well-written, though it would benefit from a thorough grammar and syntax proof. The supplemental information is confusing and of limited benefit as currently presented, and ought be substantially amended with a much clearer explanation of what is being shown, and what its implications are.

In light of the fact that the methods presented in this paper are not convincing, and that there are a number of errors related to the interpretation of data or understanding of other papers, and fundamentally that the methods are not described with sufficient clarity to be reproducible, I recommend this paper for rejection.

Major Comments:

L 116 - 126: It is not at all clear to me what is being shown here. Figure S1 appears to be crucial to the interpretation of results, but it is not in the main manuscript. The 'goodness' measure is not defined in the methods, results, or supplements, as far as I can see, nor are the seismogram vertical axes scaled or labelled. Because of this, I do not know how to validate the conclusions presented in the results section. Significantly more explanation, and linking of the text to the figures, would be extremely helpful.

L 152: As I understand it, the quality C marsquakes effectively have no meaningful phase information recorded. How do the matched filter and CC methods work in this case?

L 153 - 154: Again, it is unclear what is meant here by marsquakes being 'weak' and 'significantly affected by ambient noise'.

Temporal distribution of events and exclusion of thermal stresses: I am extremely sceptical on this point, not least because the noise levels at the lander vary enormously through the course of the day, meaning that matching any pattern when it is windy (during the day) is much harder than when it is not (during the night). How do the authors account for this?

L 189-197: It is unclear whether the authors have interpreted the Roberts et al (4) paper correctly: the M 7.5 reference is for Earth, and the Roberts paper explicitly notes that the required moment magnitude on Mars would be lower due to the reduced gravitational field strength. The Brown & Roberts is different in its conclusions, and the difference between a 7.5 and 7.9 earthquake is not trivial. Lumping these two together is confusing.

L 198 - 201: There is no convincing explanation presented for why these four events do not have any matches.

L 226: This suggests that there was a possibility an eruption would occur during the InSight window of operation. There is no real indication of this happening, though it is of course possible.

L 242: There are many reasons that a marsquake could be depleted in high-frequency energy which do not require exotic magmatic theories, such as scattering and/or intrinsic attenuation. A more convincing explanation of why these can be excluded is needed.

Fig 3: Between Sols 100 and 450, the noise levels at InSight varied enormously. How has this been accounted for? It appears to have been completely ignored which makes me doubt that these results are robust, and it is not clear how finding events 'buried' in the noise works. Figs 3A and 3B are also completely unrelated to each other, and grouping them as such is a bit illogical.

Methods section overall: If these are new methods in an InSight context, I would be much more convinced if some synthetic data were shown and analysed to illustrate that these are not false positives. I would point the authors to the InSight blind test, which provides a catalogue of known events which they might use for this purpose, or one of the associated Instaseis databases. If the same methods were applied to random noise, what are the likelihoods of matches occurring?

Minor Comments:

L 28: InSight not Insight (repeated elsewhere)

L 28: Global magnetic studies (i.e. the lack of a magnetic field) also place constraints on internal structure. If this is the topic of the second paragraph I suggest mentioning it here too.

L 34: I am not sure it makes sense to quote an average range as such. This should be clarified with a northern/southern average thickness or something similar.

L 54: Describing the Viking 2 observation as 'robust' is probably overstating it. I would suggest 'detected only one likely candidate' or something.

L 70: Details two classes of marsquakes, the rest of the paragraph details three

L 78: It is unclear what is meant by 'ambiguous' - of uncertain type?

L 83: the syntax of this sentence is erroneous. Furthermore, the absence of phases is not necessarily only a consequence of the quakes being of low magnitude. For example strong crustal scattering might eliminate all surface waves.

Manuscript title: "Repetitive marsquakes in Martian upper mantle"

Authors: Weijia Sun and Hrvoje Tkalčić

Responses to Reviewers

Reviewer #1 (Remarks to the Author):

This is an interesting manuscript proposing that marsquakes occurs in repeating way. InSight/SEIS data are furthermore not easy to understand, due to the extreme environment and the very low amplitude of both the seismic noise and observed marsquake.

We appreciate that you find our manuscript interesting.

However, the main results cannot be understood, based on simple relation between the magnitude of the proposed events, their cross-correlation coefficients with the parent's events and their expected signal to noise ratio.

We appreciate this concern, and we have worked to improve the clarity of the results by including numerous tests and new information. Equation 4 for the magnitudes of the newly detected events is an empirical relationship, far from a traditional formula used. We do not want to put too much emphasis on the magnitudes themselves. We simply offer a way to estimate their magnitudes but are open to alternative routes and so would be happy to remove the magnitudes and keep only the amplitude ratios. To make things clearer, we introduce the amplitude relationship in Equation 5. Furthermore, we examine the cross-correlation coefficients, signal-to-noise ratios, detected vs. template events amplitude ratios and their relationship with the local mean solar time in Section S3 in much greater detail.

I unfortunately suspect that the MF method is possibly corrupted by noise background, including glitches, and that the MF method is therefore more controlled by the similarities in the noise than by similarities in the quakes. This noise is indeed not a stochastic noise. It is strongly thermally controlled and is repeating, including for glitches, regularly.

Our work was inspired by the successful applications of the MF method on earthquakes, more specifically, aftershocks and volcanic events. By now, the method has been applied to detect small-magnitude repeating earthquakes buried in ambient noise. The MF detection is achieved by matching the template events (e.g., known marsquakes) in the continuous waveforms with a correlation higher than an optimal threshold.

We acknowledge glitches contaminate the Martian waveforms. Indeed, we have checked both the template and the newly detected events using synthetic and actual Martian data to ensure they are not affected by apparent glitches – a full description is given in Sections S2.2 to S2.4. The synthetic data is constructed by embedding three glitches into white noise filtered in the same frequency band as the marsquakes. The actual Martian data were the one-day-long continuous data on July 1, 2019, also carefully analyzed by Scholtz et al. (2021), who aimed to present methods of glitches removal.

We agree that there is a possibility of utilizing the matching technique to detect thermal events. Dahmen et al. (2021) reported that the thermally-controlled events are repeating, which you also mentioned. Nevertheless, these thermal events have a 5-30 Hz super high-

frequency band with a short duration (20 s). In contrast, we bandpass-filter both the template events and continuous waveforms in a band of 0.1-0.8 Hz recorded by the broadband sensors, quite different from the frequency band of thermal events. We, therefore, consider the matching of low-frequency marsquakes. We have discussed this in the first paragraph of the discussion section and Section S1.2.

Dahmen, N. L., Clinton, J. F., Ceylan, S., van Driel, M., Giardini, D., Khan, A., et al. (2021). Super High Frequency Events: A New Class of Events Recorded by the InSight Seismometers on Mars. *Journal of Geophysical Research: Planets*, 126(2), e2020JE006599.
Scholz, J. R., Widmer-Schmidrig, R., Davis, P., Lognonné, P., Pinot, B., Garcia, R. F., et al. (2020). Detection, Analysis, and Removal of Glitches From InSight's Seismic Data From Mars. *Earth and Space Science*, 7(11).

The authors must demonstrate, for example by using deglitched data, that the same results are retrieved. The authors must also ensure that the robustness of the method with respect to noise is found in figure summarizing for example the evolution of cross-correlation coefficients with the expected signal to noise of the repeating quakes.

We have checked the newly detected marsquakes to ensure they are not affected by the glitched data, which is now supported by synthetic and actual data. Please see the related tests in Sections S2.2 to S2.4.

Following your suggestions, we take one day (1 July 2019) of data contaminated with glitches and deglitched data to test the robustness of the MF method. The data was carefully analyzed by Scholz et al. (2020). We have made various robustness tests of MF, including selecting MF parameters such as the frequency band, the template length and start time, the effects of glitches, etc., in sub-sections of S2.

My recommendation is therefore to request the authors to re-submit their work with a much stronger discussion on the effects of glitches structure in their analysis, which might need to use deglitched data. In addition, the Table 1 must be associated by summary figure, which shows that a priori relations between cross-correlation coefficients, event magnitude and local time noise, are coherent with what we can expect from waveform matching of low signal to noise events. At this time, this is missing, and the reader just cannot understand the results.

Thanks for your encouragement. As explained above, the Section S2.3 now illustrates that the glitches have no influence on MF detections.

As you suggested, we carefully investigate cross-correlation coefficients, signal-to-noise ratios, detected vs. template events amplitude ratios and their relationship with the local mean solar time in Section S3. We believe that this now presents a much stronger discussion and confirms the robustness of our results.

Because I was therefore not convinced by the main results summarized in Table 1, I did not went in depth in the section describing the consequences in term of Mars tectonics.

We trust we addressed your concerns with comprehensive tests of robustness presented in Sections S2, S3 and S4.

Minor comments:

Line 15: The wording “Martian seismic data recorded by InSight are rather noisy” seems surprising. Mars noise is very significantly lower than the Earth micro-seismic noise, so with Earth standard, the Martian seismic data are very low noise. It seems to me that the issue is more that Marsquake have low magnitude and low amplitude, leading to low SNR. We agree, and we have changed the sentence accordingly.

Line 28: k2 has also provided estimate of the internal structure. See for example Smrekar et al., Pre-mission InSights on the Interior of Mars, *Space Sci Rev*, 215, 3, doi: 10.1007/s11214-018-0563-9, 2019

We now introduce their work for clarity in the text.

InSight/SEIS presentation: It is very much surprising that this paper is not citing any mission or instrument papers when it is introducing the InSight mission (line 28) nor the InSight seismometer (line 69). I suggest the authors to cite:

For the InSight mission:

Banerdt et al, Initial results from the InSight mission on Mars, *Nature Geoscience*, 13, 183-189, doi: 10.1038/s41561-020-0544-y, 2020

For the SEIS experiment:

Lognonné, P., Banerdt, W.B., Giardini, D. et al., SEIS: Insight’s Seismic Experiment for Internal Structure of Mars, *Space Sci Rev*, 215, 12, doi : 10.1007/s11214-018-0574-6 , 2019. Lognonné et al., Constraints on the shallow elastic and anelastic structure of Mars from InSight seismic data, *Nature geoscience*, 13, 213-220, doi: 10.1038/s41561-020-0536-y (2020).

Thanks. We now also cite these works in the revised manuscript.

Line 53: For the Viking 2 mission: citing a paper which in fact recite the Viking 2 mission paper is not the best way to present facts. The authors shall consider citing the Viking 2 mission paper instead

D.L. Anderson, W.F. Miller, G.V. Latham, Y. Nakamura, M.N. Toksoz, A.M. Dainty, F.K. Duennebier, A.R. Lazarewicz, R.L. Kowach, T.C. Knight, *Seismology on Mars. J. Geophys. Res.* 82, 4524–4546 (1977a) <https://doi.org/10.1029/JS082i028p04524>

We refer to the literature you suggested in the revised manuscript.

Line 104: In addition, the SEIS data are not cited when used. See guidelines in the SEIS data depositories and please cite correctly these data as: InSight Mars SEIS Data Service. (2019). SEIS raw data, Insight Mission. IGP, JPL, CNES, ETHZ, ICL, MPS, ISAE-Supaero, LPG, MFSC. https://doi.org/10.18715/SEIS.INSIGHT.XB_2016

We correct our mistake in the manuscript.

Line 106: Note that the work is made by the MarsQuake Service team, which is hosted by ETH Zurich but is not the ETH Zurich team. See for example the MQS composition in <https://doi.org/10.12686/a11>

Line 108: The same is true for the InSight Mars Quake catalogue, which must also be cited when used, with citation as: InSight Marsquake Service (2021). Mars Seismic Catalogue, InSight Mission; V6 2021-04-01. ETHZ, IPGP, JPL, ICL, MPS, Univ. Bristol. <https://doi.org/10.12686/a11>

Thanks for pointing this out; corrected.

Line 116: Fig S1, first figure shown, is poorly introducing the data. No unit are shown. I suspect that these are DU. At least some value of the DU is needed or alternatively, the data shall be plotted in physical unit, as it is mandatory to provide to the reader some idea of the amplitude of both the noise and signal, which is completely absent from this paper.

The unit of goodness is dimensionless. We now clarify this in the figure caption and insert Equation 1 in Methods.

Line 116-126: The authors use a large fraction of their main text to say why the BL approach is not working due to glitches. But they do not indicate in the main text why the MF method is working. I therefore recommend them to do the opposite. One line to say that BL is not working and 9 lines to summarize why MF is working.

We agree that the sense of the importance of the BL method in the context of our findings should be downplayed. As the glitches contaminate the data, the BL approach does not work well. The MF method based on the normalized correlation coefficients in a time window is little affected by these glitches. We moved most of the BL information to the supplementary subsection S1.2, and changed the order of summary as suggested.

Line 125: The two papers focused on InSight/SEIS glitches are missing from the reference and shall provide more background on the origin of these glitches, mostly associated to the extremely large temperature variations of on Mars (almost 100C for the tether linking SEIS to the lander for example). See Lognonné et al., 2020 cited above and Scholz et al (2020), Detection, Analysis, and Removal of Glitches From InSight's Seismic Data From Mars, Earth and Space Science, 7, e2020EA001317, doi: 10.1029/2020EA001317

Thanks for your suggestions. We introduce these two references now.

In addition, the authors shall consider to perform a deglitching of the data to ensure that glitches patterns are not corrupting their results.

As described above, we have applied the MF method on one-day deglitched data, where the glitch information was provided in Scholz et al. (2020). We illustrated the test in Section S2.3.

Line 129: while seismologists will have a look on the supplementary material, most of the reader will first read the table and will ask themselves how these results can be valid. The very tricky point are the magnitude. These magnitudes range from a minimum of 0.7 to magnitude of 4, so therefore more than the magnitude of the detected 173a and 235b. I recall that the magnitude of the detected 173a-235 b was about 3.5.

This is leading to several questions:

- What is the magnitude definition used by the authors?

- If this is a classical body wave magnitude definition, this suggest that the lowest magnitude events will have an amplitude $10^{2.8}$ lower than the 173a-235b, which means 630. The SNR of 173a and 235b are respectively about 100 and 250. Even if the background noise depends significantly on the local time, it seems not obvious that matching can provide good results for SNR much less than one, which will be what we expect for magnitude less than 1.5

- How to explain that the event S0235b-MF04, which is proposed as magnitude 4 and with correlation coefficients of 0.54 0.47 0.63, has correlation coefficients finally comparable to event S0235b-MF34 , with 0.55 0.38 0.78 for the correlation coefficients but with a magnitude of 0.7

It seems to me that these observations on the results proposed are not internally compatible. I imagine that the correlation coefficients are decreasing significantly with the magnitude and/or with the apriori signal to noise which will me roughly proportional to $10m/n$ (LMST), where n is noise as LMST.This seems not the case and therefore generate a lot of interrogation on the main message of the paper. For that reason, I believe that this analysis is not yet mature for publication.

Your concerns regarding the magnitude are well received. The idea is to introduce the magnitudes of the newly detected events using the relationship of the newly detected and template amplitudes and the reported magnitude of the corresponding template event. Equation 4 reads

$$M_{\text{detected}} = M_{\text{template}} + \mathcal{A}.$$

Note that M_{detected} is not the traditional magnitude, defined as a magnitude ratio relative to its template event. To make this definition more intuitive, we now introduce the term \mathcal{A} explicitly in Equation 5:

$$\mathcal{A} = 2 \log_{10} \left(\frac{\mathcal{A}_{MD}}{\mathcal{A}_{MT}} \right).$$

\mathcal{A}_{MD} and \mathcal{A}_{MT} represent the medians of the maximum amplitudes (in the defined time window on all three channels) of the detected and template events, respectively. Note that for the template events, the logarithm of the amplitude ratio is zero. It is positive for the amplitude ratio > 1.0 and negative for the amplitude ratio < 1.0 , according to Equation 5. Therefore, the newly detected marsquakes during the relatively noisy daylight will result in larger magnitudes (e.g., S0235b-MF04; magnitude 4.0) than the events occurring during the relatively quiet nights (e.g., S0235b-MF34; magnitude 0.7).

Although the magnitudes appear explicitly in these equations, we remove them from Table 1 and instead provide only amplitude ratios.

Reviewer #2 (Remarks to the Author):

This paper is describing a method to find Martian quakes occurring at the same place than other martian quakes already detected by InSight mission.

This method is applied to the continuous recordings of SEIS instruments, and the authors claim to have found a large number of quakes identical to the two largest seismic events recorded by SEIS instrument.

Before and after the presentation of the main results, the author elaborate what are the implications of such repeated signals, interpreted as Mars Quakes, for Mars tectonics, volcanism and internal dynamics.

I will not review here the implications of these results for Mars internal dynamics.

The main reason being that the strong weak point of the paper is the reliability of these results (repeating signals on SEIS seismometer).

Thank you for your constructive review. We revise the manuscript and reinforce our initial results by performing various comprehensive tests of the MF method robustness in Sections S2, S3 and S4. More specifically, the synthetic experiments that guided the selection of the MF method parameters are given in Section S2. We further investigate cross-correlation coefficients, signal-to-noise ratios, detected vs. template events' amplitude ratios and their relationship with the local mean solar time in Section S3. Finally, Section S4 shows an analysis supporting our interpretation of the repetitive marsquakes. We respond to your concerns point-by-point below.

First, the method presented here contains some missing information that do not allow to reproduce the results, and some arbitrary parameter choices that strongly influence the results. The threshold used to validate a detection is set arbitrarily to 7 times the "MAD". Why seven? how do justify it? Probably better to use a statistical argument based on distribution of MAD. This threshold appears to vary significantly on figures provided as supplementary material. Why? Is it due to the varying size of the window used to compute the MAD? please provide the size of this window to ensure that the results can be reproduced.

We acknowledge the importance of parameter choices, which is universally true for all data processing or analyses. We comprehensively evaluate these parameters in the updated supplementary materials to address your concerns and make our choices transparent, i.e., Section S2. We consider a threshold such that the noise (including the glitches) should not be detected as a possible event match. More specifically, the selection of the threshold is given in Section S2.2.

Furthermore, the threshold is dynamic. The cross-correlation coefficient (CC) is varied as we run the template matching over the 24-hour continuous Martian data. The median absolute deviation (MAD) is calculated from CC, which leads to the variation of MAD.

We now describe the workflow and parameter selection in Section S1, ensuring that the results can be reproduced. We also provide the MF code and the test data.

Second, the robustness of the detection method is questionable for the following reasons. No synthetic examples are provided. Robustness of the detection is not tested for different window sizes and/or different S wave arrival times. Most the detections based on S wave

are providing very low correlation coefficients for P waveform comparison. Can you show how the P amplitude (by scaling the reference waveforms) would compare to signals assumed as P? Compute an SNR? If the method is working, you would expect that the stack of the waveforms would converge to the waveform of the reference event, like it was done for the deep Moon quakes. This simple test is not even tried by the authors.

We now test the robustness of the MF method in Section S2 comprehensively, including tests on synthetic noise shown in Fig. S2.5 and actual Martian data as you suggested. The comprehensive tests include different window sizes and the start time of the template, in which the S arrival time is fetched from the extended data of Giardini et al. (2020). The length of the time window is given as twice longer than the longest period (it is 10 s, i.e., 0.1 Hz in our case). The selection of template length and start time is scrutinized in Sections S2.5 and S2.6. Considering phase-arrival uncertainties as illustrated in Giardini et al. (2020), also provided below, we eventually take a 22s-long window from -2 s to 20 s relative to the S-wave arrival time listed in the figure below.

Our detection is based on the S-wave arrival, as you pointed out. It is no surprise that the P phases usually show low correlation coefficients due to their lower SNR than the SNR of S phases. This is because S-wave arrivals have a much larger amplitude and are radiated in a broader frequency range than their P-wave counterparts (Clinton et al., 2021), which may mean that the P phases are more affected by noise than S phases. In addition, the P-wave arrival of S0173a is contaminated by a glitch. We further investigate the cross-correlation coefficients, signal-to-noise ratios, detected vs. template events amplitude ratios and their relationship with the local mean solar time in Section S3.

We now stack the newly detected events for a comparison with the reference event shown below. The stacked waveforms have good cross-correlation coefficients with the template waveforms. Thank you for your suggestions.

ARTICLES		NATURE GEOSCIENCE					
Event name	Event quality	Aligned distance (°)	Catalogue distance (°)	Catalogue back-azimuth (°)	P pick (s) Time (uncertainty)	S pick (s) Time (uncertainty)	Magnitude M _w
Low Frequency events							
S0105a	B	27 (±5)	31 (±10)		2019-03-14 21:03:31 (±20)	2019-03-14 21:06:39 (±20)	3.2
S0133a	B	[90] (±20)	33 (±7)		2019-04-12 18:14:35 (±60)	2019-04-12 18:17:56 (±20)	3.2
S0154a	B	[90] (±20)	49 (±7)		2019-05-04 07:07:05 (±20)	2019-05-04 07:11:57 (±20)	3.5
S0167a	C	[150] (±20)					3.8
S0173a	A	28 (±3)	29 (±3)	91 (±5)	2019-05-23 02:22:59.1 (±1)	2019-05-23 02:25:53.8 (±2)	3.6
S0183a	C	47 (±10)		73 (±20)	2019-06-03 02:27:45.8 (±1)	2019-06-03 02:32:09 (±10)	3.1
S0185a	B	60 (±3)	59 (±10)		2019-06-05 02:13:51 (±20)	2019-06-05 02:19:35 (±20)	3.1
S0189a	B	27 (±5)	32 (±7)		2019-06-09 05:40:06 (±20)	2019-06-09 05:43:20 (±20)	3.0
S0205a	D	[45] (±10)					3.0
S0226b	C	[90] (±20)					
S0234c	D	65 (±5)					2.8
S0235b	A	25 (±3)	26 (±3)	74 (±15)	2019-07-26 12:19:19.3 (±2)	2019-07-26 12:21:56.1 (±2)	3.6
S0325a	B	35 (±5)	40 (±5)		2019-10-26 06:58:58.9 (±1)	2019-10-26 07:02:56 (±10)	3.7

Figure from Giardini et al. (2020).

Giardini, D., Lognonné, P., Banerdt, W. B., Pike, W. T., Christensen, U., Ceylan, S., et al. (2020). The seismicity of Mars. *Nature Geoscience*, 13(3), 205-212.

Clinton, J. F., Ceylan, S., van Driel, M., Giardini, D., Stähler, S. C., Böse, M., et al. (2021).

The Marsquake catalogue from InSight, sols 0–478. *Physics of the Earth and Planetary Interiors*, 310, 106595.

Figure. Stacked waveforms of MF detections for the template of A) S0173a and B) S0235b. The red lines denote the template of S0173a and S0235b, while the black lines are the stacked waveforms. The cross-correlation coefficients are labelled in the top left corner.

Third, there is some wrong interpretation of the input data and the results of the method. The frequency range used here (0.1-0.8 Hz) is going to too low frequencies, in a region (0.1-0.2 Hz) presenting much more noise than above 0.3 Hz. This noise is driven by a combination of Pressure, Wind effects and glitches that can repeat due to atmospheric forcing and temperature periodicity of glitches. Due to the S wave pick for S0235b being too early by more than 5s, the long period noise before the S arrival, but in the S window used, is dominating the correlation coefficient estimates. This is the reason why so many quakes are detected for that particular event (that particular S waveform). In addition, using the inclined U,V,W components, and not ZNE, give more weight to the tilt noise sources expressing mainly along the horizontal components below 0.3 Hz.

The frequency is indeed delicate. Clinton et al. (2021) stated that the LF family events are dominated in the band of 0.1-1 Hz (1-10 s) in their introduction section, while Scholz et al. (2020) performed a bandpass filter of 0.1-0.8 Hz for receiver function imaging as provided in their Figure 8 caption. With some noise considerations, we were guided by these choices for our frequency band of 0.1-0.8 Hz. Garcia et al. (2020) investigated the behavior of pressure signals in detail. They showed that the low-frequency seismic records agree with the atmospheric pressure in the frequency range 0.03-0.5 Hz during the day (see Figure 1b in that paper, also shown below).

We also perform a time-frequency analysis over the continuous waveform containing substantial amplitude variations from wind and temperature, similar to the study of Scholz et al. (2020). From this analysis, the signals and noise have an overlapping frequency band, i.e., 0.1-0.3 Hz, and the potential marsquakes could be buried in the ambient noise.

According to your suggestion, we now make a similar analysis for the frequency band of

0.1-0.3 Hz in Section S2.1. After these tests, we finally apply a broad frequency band of 0.1-0.8 Hz to bandpass-filter low-frequency marsquakes and continuous waveforms. The frequency band can effectively suppress ambient noise during the Martian day, as seen in Figures S3.9 and S3.10.

We calculated the incident angle of the S0173a, S0183a and S0235b marsquakes located beneath the InSight SEIS. The incident angle is about 27° , which approaches the oblique sensors ($\sim 30^\circ$). In addition, the horizontal NE components have more agreement with pressure perturbations than the vertical Z components (Garcia et al., 2020). With these considerations, we apply the MF methods to the UVW components. We have mentioned this in supplementary text S1.1.

Figure from Garcia et al. (2020). It shows that the pressure signal has the highest energy in the time interval 11-13h.

Clinton, J. F., Ceylan, S., van Driel, M., Giardini, D., Stähler, S. C., Böse, M., et al. (2021). The Marsquake catalogue from InSight, sols 0–478. *Physics of the Earth and Planetary Interiors*, 310, 106595.

Scholz, J. R., Widmer-Schmidrig, R., Davis, P., Lognonné, P., Pinot, B., Garcia, R. F., et al. (2020). Detection, Analysis, and Removal of Glitches from InSight's Seismic Data From Mars. *Earth and Space Science*, 7(11).

Garcia, R. F., Kenda, B., Kawamura, T., Spiga, A., Murdoch, N., Lognonné, P. H., et al. (2020). Pressure Effects on the SEIS-InSight Instrument, Improvement of Seismic Records, and Characterization of Long Period Atmospheric Waves from Ground Displacements. *Journal of Geophysical Research: Planets*, 125(7).

Concerning the results, events S0173a-MF01 and S0173a-MF02 correspond to the SS and SSS multiples of the S wave of S0173a. S-P, SS-S (S0173a-MF01 minus S0173a) and SSS-S (S0173a-MF02 minus S0173a) differential of this event can be consistently explained by ray tracing in Mars internal structure models.

With the aid of various Martian models from Khan et al. (2016), we can calculate theoretical arrival times for different seismic phases. Although these models should be taken with a

big grain of salt, the differential travel times: S-P, SS-S and SSS-S are reasonable candidates for interpreting S0173a-MF01 and S0173a-MF02. As the delays of SS and SSS vary significantly for different Martian models, we cannot rule out the possibility of these two detections being the later arrivals of S0173a. We discuss this in Section S4.

Khan, A., van Driel, M., Böse, M., Giardini, D., Ceylan, S., Yan, J., et al. (2016). Single-station and single-event marsquake location and inversion for structure using synthetic Martian waveforms. *Physics of the Earth and Planetary Interiors*, 258, 28-42.

Finally, the authors observe moment magnitudes larger during day time than during night time. This observation can be easily explained if you assume that the method is fitting noise, because the amplitude of the noise is much larger during the day.

We conducted comprehensive tests of the MF method, confirming that it does not fit the noise but the signals from marsquakes. Please refer to Section S2.

Note that the formula for a matched event magnitude contains the amplitude ratio between the newly detected and the template event. Therefore, the magnitude listed in Table 1 is not traditional but instead defined relative to the reference events. To clarify this, we explicitly express the amplitude relationship between the detected and template events in Equation 5. We comprehensively investigate cross-correlation coefficients, signal-to-noise ratios, detected vs. template events amplitude ratios and their relationship with the local mean solar time in Section S3. The amplitude ratios for most newly detected events are smaller than those of the template events during the day and night, as shown in Figure S3.8.

Even if the method and results are not convincing, I would encourage the authors to pursue in their studies using match filtering, targeting on the resolution of the above mentioned issues.

We have taken an opportunity to revise our manuscript in response to your and other reviewers' comments and have hopefully succeeded in putting away many of your concerns. We initially wanted to keep the manuscript concise, but we are convinced that the revamped, extensive supplementary material works better. The supplements now contain new sections, and of particular importance for addressing your concerns are Sections S2-4.

Additional comments:

- the InSight catalog is produced by Mars Quake Service (in which at least 5 different organisations are involved) not by ETHZ team.

We have corrected this reference. Thank you!

Reviewer #3 (Remarks to the Author):

Summary: The authors apply two signal processing techniques to identify repetitive marsquakes in InSight data. Using Benford's Law-based statistics and matched filter templates, they report detection of 47 new seismic events from the Cerberus Fossae region of Mars. From this, they conclude that the martian mantle is still mobile and seismically highly active.

Overall comments: Benford's Law and matched filter banks are not, as far as I know, standard techniques currently used to identify marsquakes in InSight data. Therefore, if they can be shown to robustly identify events and exclude false positives, they have significant potential in terms of application to martian seismic data.

Indeed, as far as we know, the two techniques have not yet been used to detect marsquakes by other groups or individual researchers. Employing these methods is the main innovative aspect of our manuscript. The methods are transferrable to other planetary studies. Various researchers succeeded in implementing them in earthquake detections, as explained in the introduction. More specifically, the matched-filter method is now a standard method for tracking earthquake evolution and detecting volcanic activities.

To address this and other reviewers' comments, we make significant efforts in this revision to show the credibility and robustness of the MF method in Section S2. These comprehensive tests include the following: S2.1 The choice of the frequency band, S2.2 The threshold of matched-filter detection, S2.3 The effect of glitches on MF detections, S2.4 Tests of events buried in continuous noise, S2.5 The effect of the template length, and S2.6 The effect of the template start time.

We further investigate cross-correlation coefficients, signal-to-noise ratios, detected vs. template events amplitude ratios and their relationship with the local mean solar time in Section S3. Finally, Section S4 shows an analysis supporting our interpretation of the repetitive marsquakes.

However, at present it is unclear whether the results presented herein are robust and reliable. Some of the conclusions I find extremely surprising, especially with regard to the lack of diurnal patterns in the detected marsquakes. Whilst this does not mean that the techniques are incorrectly applied or invalid, it does mean that they are not thoroughly presented enough to be convincing.

We admit that we had underestimated the importance of presenting extensive supplementary materials with the initial version of the manuscript. We, therefore, take this opportunity to reinforce our manuscript by introducing all-embracing tests in Sections S2, S3 and S4, as mentioned above. As for the comment on the surprising aspect, we think that the reviewer will agree that there are many surprising aspects of the InSight mission results, and the surprise element is not necessarily negative.

In terms of structure, the manuscript is reasonably logical and well-written, though it would benefit from a thorough grammar and syntax proof. The supplemental information is confusing and of limited benefit as currently presented, and ought be substantially amended with a much clearer explanation of what is being shown, and what its implications are.

The reviewer can rest assured that this has been already done for both the original

manuscript and this revision. We agree that we provided limited supplemental information, and from that point of view, this revision has been a chance to redeem ourselves. We now extend the supplementary materials into four sections as mentioned above, in which each subsection is logically organized.

In light of the fact that the methods presented in this paper are not convincing, and that there are a number of errors related to the interpretation of data or understanding of other papers, and fundamentally that the methods are not described with sufficient clarity to be reproducible, I recommend this paper for rejection.

Perhaps we did ourselves a disservice by not presenting enough supplementary material in the originally submitted paper, but we disagree that the methods are not convincing. In light of the reviewer's comments and the newly published articles on Martian structures that all use extensive supplementary components, we changed our approach here and provided extensive new information.

We have put significant effort into the method robustness tests in expanding Sections S2, S3 and S4. We believe that the methods are described with sufficient clarity to be reproducible. More specifically, we now summarize the workflow and parameter setting in Section S1.1 to ensure our detections are reproducible. We also make related codes available.

We have correctly understood and interpreted Robert's results of the magnitude of paleomarsquakes. Please see our responses on that part in one of the comments below.

Major Comments:

L 116 - 126: It is not at all clear to me what is being shown here. Figure S1 appears to be crucial to the interpretation of results, but it is not in the main manuscript. The 'goodness' measure is not defined in the methods, results, or supplements, as far as I can see, nor are the seismogram vertical axes scaled or labelled. Because of this, I do not know how to validate the conclusions presented in the results section. Significantly more explanation, and linking of the text to the figures, would be extremely helpful.

We are grateful to the reviewer for pointing this out. We now provide the goodness definition (Equation 1) in the Methods section. We conduct comprehensive tests in Sections S2, S3 and S4 to support our conclusions.

L 152: As I understand it, the quality C marsquakes effectively have no meaningful phase information recorded. How do the matched filter and CC methods work in this case?

We do not use quality C marsquakes. We choose nine marsquakes as templates and detect 47 new events. The four template marsquakes for which we actually detect matching events are S0173a (quality A), S0235b (quality A), S0183a (quality B), and S0325a (quality B) listed in Table 1.

The reviewer may wonder about the marsquake S0183a, listed as the quality C in Table S1. The quality C for that event originally came from Extended data Fig 4 of Giardini et al. (2020), which may be a typo. However, the S0183a quality has been fixed to B in the updated marsquake catalogue v6. In conclusion, we use only marsquakes of quality A and B.

Giardini, D., Lognonné, P., Banerdt, W. B., Pike, W. T., Christensen, U., Ceylan, S., et al. (2020). The seismicity of Mars. *Nature Geoscience*, 13(3), 205-212.

L 153 - 154: Again, it is unclear what is meant here by marsquakes being ‘weak’ and ‘significantly affected by ambient noise’.

We now rephrase that to ‘This might mean that the signal-to-noise ratio of template marsquakes are relatively low as they could be significantly affected by ambient noise.’.

Temporal distribution of events and exclusion of thermal stresses: I am extremely sceptical on this point, not least because the noise levels at the lander vary enormously through the course of the day, meaning that matching any pattern when it is windy (during the day) is much harder than when it is not (during the night). How do the authors account for this?

We are familiar with the report of the repeating thermally-controlled events by Dahmen et al. (2021). Fortunately, these thermal events are quite different from the low-frequency events we pursue in this work. These thermal events have a 5-30 Hz super high-frequency band with a short duration (20 s) around sunset. In contrast, we bandpass-filter both the template events and continuous waveforms in a band of 0.1-0.8 Hz, quite different from the frequency band of thermal events. In addition, our detections do not show a clear time cluster. We discuss this in the first paragraph of the Discussions section and Section S1.2. We analyze the spectrogram of an entire Martian day in Section S2.1 and the absolute amplitudes with/without bandpass filter in Figs. S3.9 and S3.10. These analyses indicate that most of the strong wind and pressure noise could be effectively suppressed. In addition, the wind is not continuous but intermittent. Our newly detected events during the day are located in the time intervals with smaller absolute amplitudes, as shown in Figs. S3.9E and S3.10E.

Dahmen, N. L., Clinton, J. F., Ceylan, S., van Driel, M., Giardini, D., Khan, A., et al. (2021). Super High Frequency Events: A New Class of Events Recorded by the InSight Seismometers on Mars. *Journal of Geophysical Research: Planets*, 126(2), e2020JE006599.

L 189-197: It is unclear whether the authors have interpreted the Roberts et al (4) paper correctly: the M 7.5 reference is for Earth, and the Roberts paper explicitly notes that the required moment magnitude on Mars would be lower due to the reduced gravitational field strength. The Brown & Roberts is different in its conclusions, and the difference between a 7.5 and 7.9 earthquake is not trivial. Lumping these two together is confusing.

We cite the two references to refer to the magnitude of paleomarsquakes. This has minor relevance to our results. However, given that the reviewer raised this point, we offer the following explanation: Roberts et al. (2012) discussed the magnitude “Thus, if we push this interpretation to its limit, the best estimate we can make of paleomarsquake magnitude may be best led by the boulder trail densities which show anomalously high boulder trail densities for an along-strike distance of ~150–200 km (~3–4° of longitude). **If we assume this was produced by a single paleomarsquake**, which we stress we have not proven and is an assumption, **then the implied magnitude is large**. On Earth, the earthquake

magnitude needed to produce an area of ground shaking that is 150 km across that is severe enough to mobilize meter-scale boulders would involve a slip patch of similar dimensions, **so a magnitude of ~7.5 Mw would be implied** [Wells and Coppersmith, 1994].” This could be ambiguous.

In a newer paper, Brown and Roberts (2019) wrote, “**The ~207-km-wide zone of mobilized boulders measured along Cerberus Fossae might be consistent with a marsquake of moment magnitude ~M7.9** (see Wells & Coppersmith, 1994).”

Therefore, we think we correctly interpreted the discussion of the two of Roberts’s papers.

Brown, J. R., & Roberts, G. P. (2019). Possible Evidence for Variation in Magnitude for Marsquakes From Fallen Boulder Populations, Grjota Valles, Mars. *Journal of Geophysical Research: Planets*, 124(3), 801-822.

Roberts, G. P., Matthews, B., Bristow, C., Guerrieri, L., & Vetterlein, J. (2012). Possible evidence of paleomarsquakes from fallen boulder populations, Cerberus Fossae, Mars. *Journal of Geophysical Research: Planets*, 117(E2).

L 198 - 201: There is no convincing explanation presented for why these four events do not have any matches.

We do not know why. However, it is probably because of the large uncertainties of S arrival times for these events. The reviewer can refer to the information provided by Giardini et al. (2020). For these four events, the S arrival time error is up to ± 20 s. This is in contrast to the events with smaller arrival time uncertainties, for which we succeed in detecting matching events.

ARTICLES				NATURE GEOSCIENCE			
Event name	Event quality	Aligned distance (°)	Catalogue distance (°)	Catalogue back-azimuth (°)	P pick (s) Time (uncertainty)	S pick (s) Time (uncertainty)	Magnitude M _w
Low Frequency events							
S0105a	B	27 (±5)	31 (±10)		2019-03-14 21:03:31 (±20)	2019-03-14 21:06:39 (±20)	3.2
S0133a	B	[90] (±20)	33 (±7)		2019-04-12 18:14:35 (±60)	2019-04-12 18:17:56 (±20)	3.2
S0154a	B	[90] (±20)	49 (±7)		2019-05-04 07:07:05 (±20)	2019-05-04 07:11:57 (±20)	3.5
S0167a	C	[150] (±20)					3.8
S0173a	A	28 (±3)	29 (±3)	91 (±5)	2019-05-23 02:22:59.1 (±1)	2019-05-23 02:25:53.8 (±2)	3.6
S0183a	C	47 (±10)		73 (±20)	2019-06-03 02:27:45.8 (±1)	2019-06-03 02:32:09 (±10)	3.1
S0185a	B	60 (±3)	59 (±10)		2019-06-05 02:13:51 (±20)	2019-06-05 02:19:35 (±20)	3.1
S0189a	B	27 (±5)	32 (±7)		2019-06-09 05:40:06 (±20)	2019-06-09 05:43:20 (±20)	3.0
S0205a	D	[45] (±10)					3.0
S0226b	C	[90] (±20)					
S0234c	D	65 (±5)					2.8
S0235b	A	25 (±3)	26 (±3)	74 (±15)	2019-07-26 12:19:19.3 (±2)	2019-07-26 12:21:56.1 (±2)	3.6
S0325a	B	35 (±5)	40 (±5)		2019-10-26 06:58:58.9 (±1)	2019-10-26 07:02:56 (±10)	3.7

Giardini, D., Lognonné, P., Banerdt, W. B., Pike, W. T., Christensen, U., Ceylan, S., et al. (2020). The seismicity of Mars. *Nature Geoscience*, 13(3), 205-212.

L 226: This suggests that there was a possibility an eruption would occur during the InSight window of operation. There is no real indication of this happening, though it is of course possible.

We rephrase the sentence to avoid misunderstanding, ‘An eruption did not occur on Mars

during the InSight project; therefore, the volcanic activity could be due to the prevalent intrusive magmatism.’.

L 242: There are many reasons that a marsquake could be depleted in high-frequency energy, which do not require exotic magmatic theories, such as scattering and/or intrinsic attenuation. A more convincing explanation of why these can be excluded is needed.

The paragraph first stated that volcanic events that have recently been discovered are deep, low-frequency earthquakes that occur at quiescent volcanoes due to the repeated pressurization of volatiles in subcrustal magma on Earth. This observation shares at least one common characteristic with the Cerberus Fossae region marsquakes detected here – they are rich in low-frequency content.

We now replace ‘depleted in high-frequency content’ with ‘rich in low-frequency content’ to avoid confusion.

Fig 3: Between Sols 100 and 450, the noise levels at InSight varied enormously. How has this been accounted for? It appears to have been completely ignored which makes me doubt that these results are robust, and it is not clear how finding events ‘buried’ in the noise works. Figs 3A and 3B are also completely unrelated to each other, and grouping them as such is a bit illogical.

We made comprehensive tests to illustrate the robustness of the MF method in Sections S2 and S3. We test the detections of two known marsquakes: S0173a and S0235b buried in synthetic noise. The S arrival amplitudes are normalized to 0.5, 0.75 and 1.0 times of noise amplitude. We have successful self-detections of the two events. The test is shown in Section S2.4 and Figs. S2.12 to S2.20. Moreover, with analyses of the spectrogram in Fig. S2.1 and Figs. S3.9 and S3.10, most of the strong wind and pressure noise could be effectively suppressed. In addition, the wind may not be continuous but intermittent. This may be inferred from the fact that our newly detected events during the day are located in the time intervals with smaller absolute amplitudes, as shown in Figs. S3.9E and S3.10E. We now split Figs 3A and 3B in the updated text.

Methods section overall: If these are new methods in an InSight context, I would be much more convinced if some synthetic data were shown and analysed to illustrate that these are not false positives. I would point the authors to the InSight blind test, which provides a catalogue of known events which they might use for this purpose, or one of the associated Instaseis databases. If the same methods were applied to random noise, what are the likelihoods of matches occurring?

We are thankful for these suggestions. We introduce the noise data test, i.e., random noise after the same bandpass filter and glitches, to test the method's robustness given in Section S2. Please refer to Sections S2, S3 and S4 for other relevant tests.

Minor Comments:

L 28: InSight not Insight (repeated elsewhere)

We fixed these terms throughout the manuscript.

L 28: Global magnetic studies (i.e. the lack of a magnetic field) also place constraints on internal structure. If this is the topic of the second paragraph I suggest mentioning it here too.

We mention this in the first paragraph and introduce a reference.

L 34: I am not sure it makes sense to quote an average range as such. This should be clarified with a northern/southern average thickness or something similar.

We aimed to provide background information. We rephrase the sentence.

L 54: Describing the Viking 2 observation as ‘robust’ is probably overstating it. I would suggest ‘detected only one likely candidate’ or somesuch.

We change the statement following the reviewer’s suggestion.

L 70: Details two classes of marsquakes, the rest of the paragraph details three

We re-arranged the structure of the paragraph.

L 78: It unclear what is meant by ‘ambiguous’ - of uncertain type?

We here mean low signal-to-noise ratio. For clarity, we replace the ‘ambiguous’ with ‘weak’.

L 83: the syntax of this sentence is erroneous. Furthermore, the absence of phases is not necessarily only a consequence of the quakes being of low magnitude. For example strong crustal scattering might eliminate all surface waves.

We rephrase the expression of this sentence as “Due to the relatively small magnitude of marsquakes, the uncertainties in the arrivals of identifiable phases are significant, which leads to difficulties in locating most marsquakes”.

In summary, we much appreciate the time and efforts of the reviewer, which helped us improve the quality of the manuscript.

REVIEWER COMMENTS

Reviewer #3 (Remarks to the Author):

Abstract:

The abstract is now much more clearly phrased, though I would be clear about what is meant by 'tidal modulation' - presumably tides induced by Phobos (also in L 180). The first sentence of the abstract is poorly written and needs rephrasing in a grammatically correct manner.

Main text:

L 32: astronomical considerations - geodetic perhaps a better word?

L 37: I would suggest making clear why there are two different non-coincident measurements

L 67: It might be worth outlining what the consequences for marsquake detectability would be if the linear features were from said extensive forces, as this sentence is a bit disjointed in places.

L 73: SEIS should be introduced as part of the InSight apparatus

L 78: Again, L 74 introduces two types of events whilst a third kind is mentioned in L 78. This is confusing.

L 93: Would MF and BL not still be algorithms, of a type? If so L 93 might need to be reworded.

L 112: 'which is possible for... events' does not make sense

L 121: This paragraph is much more convincing now than as previously worded.

L 148: The issue of reverberations seems like it needs to be discussed further.

L 173: This is a valuable part of the discussion, but should be better integrated to the rest of the paper through one or two introductory sentences. . To a non-specialist, it may not be obvious why moonquakes are a useful or relevant comparison.

L 178: are direct comparisons of magnitudes across different bodies meaningful? I am not sure?

L 203: why seven? I am still not sure as to why this threshold has been chosen

L 214: 'the sheer existence...evolutions' - I am not sure what is meant here? That the presence of aftershocks indicates that the system is still evolving in time relatively dynamically?

L 244: I still believe that a direct, unqualified comparison of frequency content is potentially misleading, as it suggests (without clearly stating) that the similar low-frequency richness is indicative of volcanic activity. I suggest inserting some qualification to this sentence, such as 'though this does not necessarily indicate...'

L 247: this paragraph appears to contain a number of important details, but it should be clearer how it all ties together. For example, how does the distance to Elysium Mons relate to the ages of zircons - presumably this is linked to the ongoing activity of the mantle, but it ought be clearer.

Methods:

L 490: A substantial amount of the analysis from here onward appears to rely on the assumption that seismic noise does not follow a BL distribution due to the decreased dynamic range. How carefully has this assumption been tested? For example, if the data are filtered in the 0.2-0.9Hz range as per other papers, does this noticeably change the BL distribution of noise?

L 496: would it be fairer (and clearer) to instead phrase this as 'the ground motion time series during the two marsquakes analysed follows a BL distribution'? Because 'confirm' suggests that they were known a priori to be BL distributed, which as far as I know is not known for Mars.

L 529: is this true for all events or just 0173a? This is confusing.

As per the response to the first reviewer, it is not clear to me how the authors can be sure that the strong seismic noise is not contaminating the MF results? Especially given the strong diurnal variation

in this noise, I would expect some influence upon the likelihood of matching a template event which occurred during the quiet part of the day (as the largest quakes are) to the noisiest part of the day where new events are identified.

Overall comments:

This manuscript is now more clearly presented, and I at least understand the basis of the methodology that the authors are applying. I must admit that I am still unconvinced about the results, however - not least because there are number of questions I had that are still unresolved. For example (summarising some and repeating from my previous comments, in no particular order):

What is the impact of the noise on the MF method, and how is this robustly explored and quantified? What is the authors' interpretation of the diurnal patterning (or lack thereof) in matched events? How does this compare with previous papers, and how is this discrepancy resolved with a satisfactory explanation?

Does BL actually 'detect hitherto undetected events' (as per the abstract), or is it only used to 'confirm the main low-frequency effects (in the methods section)? If it is only the latter, I wonder whether it needs to be included in the paper in such a detailed manner? The space might be better used exploring the questions raised above about the MF method.

My previous comments about it being surprising that some events are repeating whilst others are not still stands: this does not necessarily mean that there is a mistake in the workflow, but can any reasonable suggestion or speculation be made? Of course, not every method will work for every event, but without understanding why this is it is harder to accept that this method is reliable and robust. I do not quite follow the suggestion about 'suppression' of wind noise.

The supplied code is not in a usable format - I realise that this is not formally to be reviewed, but I opened it to try and follow the workflow and the README was not sufficient to know what was going on.

Reviewer #4 (Remarks to the Author):

Using the Matched Filter method and Benford's Law, this study attempted to detect previously undetected events from seismic data recorded by the InSight seismometer SEIS. The authors reported 47 new seismic events, most of which are attributed to repetitive quakes with two previously detected events. Given the low signal-to-noise ratio of the seismic data, the authors' effort to extract useful signals from the raw data is very valuable and promising. The techniques used in this study may have significant potential in applying to InSight seismic data and other planetary seismic data in future. Also, if the newly detected events are robust and reliable, the new results have significant implications for the internal dynamics and evolution of Mars.

The major concern raised by other reviewers for the original manuscript is the reliability and robustness of the methods and results. In the revised manuscript, the authors have made comprehensive tests to reinforce the reliability and robustness of the results. Therefore, I would recommend the paper can be accepted for publication.

I only have some minor suggestions:

Line32: "astronomical considerations" should be more specific. I think the major constraint in ref(7) is tidal response, i.e. Love number k_2 .

Line50: Discussion of the Martian dynamo and core state should refer to "Stevenson, D. J. (2001). Mars' core and magnetism. *Nature*, 412(6843), 214–219."

Line51: Again, "the satellite data" should be more specific here.

Line98: Although two methods are detailed in the Method section, it would be helpful to briefly

describe the principle of the methods in the introduction.

Line103: "the same general location" is a bit vague to me.

Manuscript title: "Repetitive marsquakes in Martian upper mantle"

Authors: Weijia Sun and Hrvoje Tkalčić

January 15, 2022

Responses to Reviewers

Reviewer #3 (Remarks to the Author):

Abstract:

The abstract is now much more clearly phrased, though I would be clear about what is meant by ‘tidal modulation’ - presumably tides induced by Phobos (also in L 180). The first sentence of the abstract is poorly written and needs rephrasing in a grammatically correct manner.

We tried rephrasing the first sentence with the following version: ‘Marsquakes excite seismic wavefield, allowing the Martian interior structures to be probed.’ Moreover, we attempted to write our entire manuscript clearly and coherently.

Yes, Phobos imposes a tidal force on Mars (e.g., Manga et al., 2019). We supplement the information in the abstract (Line 16) and the main text (Lines 172-173).

Manga, M., Zhai, G., & Wang, C. Y. (2019). Squeezing marsquakes out of groundwater. Geophysical Research Letters, 46(12), 6333-6340.

Main text:

L 32: astronomical considerations - geodetic perhaps a better word?

‘astronomical’ is changed to ‘geodetic’ (now in line 29).

L 37: I would suggest making clear why there are two different non-coincident measurements

The measurements stem from limited seismic constraints based on several marsquakes with a relatively high signal-to-noise ratio. We now mention this in lines 34-35.

L 67: It might be worth outlining what the consequences for marsquake detectability would be if the linear features were from said extensive forces, as this sentence is a bit disjointed in places.

We modified the sentence slightly to make the context clearer, and the last sentence in this paragraph emphasizes the consequences of marsquakes detectability (Lines 65-66).

L 73: SEIS should be introduced as part of the InSight apparatus

Thank you. We now mention SEIS as a part of the InSight apparatus in the text (Line 72).

L 78: Again, L 74 introduces two types of events whilst a third kind is mentioned in L 78.

This is confusing.

We now re-organize the text slightly to unambiguously introduce the types of events (Lines 75-77).

L 93: Would MF and BL not still be algorithms, of a type? If so L 93 might need to be reworded.

MF is a method, BL is a law, and multiple algorithms exist to analyze data in their frameworks. We here mean traditional automated algorithms like STA/LTA. We now clarify this in line 92.

L 112: 'which is possible for... events' does not make sense

We remove that part of the sentence.

L 121: This paragraph is much more convincing now than as previously worded.

Thanks for this assessment.

L 148: The issue of reverberations seems like it needs to be discussed further.

We dedicate more attention to the reverberation issue in Section S4. The issue stems mainly from the significant uncertainties of the current Martian structural models. We mention this in the main text (Lines 146-148) and refer to Section S4.

L 173: This is a valuable part of the discussion, but should be better integrated to the rest of the paper through one or two introductory sentences. . To a non-specialist, it may not be obvious why moonquakes are a useful or relevant comparison.

Thanks. We now introduce the matter in lines 171-172.

L 178: are direct comparisons of magnitudes across different bodies meaningful? I am not sure?

A direct comparison of magnitudes is not meaningful, as you point out. The magnitude information is simply provided for informative purposes.

L 203: why seven? I am still not sure as to why this threshold has been chosen

We consider a threshold that will prevent the noise (including the glitches) from being falsely detected as a possible event match. We evaluate the selection of threshold via synthetic and Martian noise data. More specifically, the choice of the threshold is described in Section S2.2.

Furthermore, the threshold is dynamic. The cross-correlation coefficient (CC) is varied as we run the template matching over the 24-hour continuous Martian data. The median absolute deviation (MAD) is calculated from CC, which leads to the variation of MAD.

We describe the workflow and parameter selection in Section S1, ensuring that the results can be reproduced. We also provide more detailed information on the MF code and the test data in the updated README file.

L 214: 'the sheer existence...evolutions' - I am not sure what is meant here? That the

presence of aftershocks indicates that the system is still evolving in time relatively dynamically?

To avoid confusion, we restate the sentence to ‘The mere existence of aftershocks (with or without high waveform similarities) suggests continuous or intermittent stress accumulation and release.’ (Lines 214-216)

L 244: I still believe that a direct, unqualified comparison of frequency content is potentially misleading, as it suggests (without clearly stating) that the similar low-frequency richness is indicative of volcanic activity. I suggest inserting some qualification to this sentence, such as ‘though this does not necessarily indicate...’

OK. We have inserted this statement (Lines 244-246).

L 247: this paragraph appears to contain a number of important details, but it should be clearer how it all ties together. For example, how does the distance to Elysium Mons relate to the ages of zircons - presumably this is linked to the ongoing activity of the mantle, but it ought be clearer.

We now rework the above in lines 257-258. In the original paper of Costa et al. (2020) reporting the age of Martian meteorites, it is considered that the young zircons were from the Tharsis or Elysium volcanic provinces. This indicates that the meteorite has a fair probability of originating from Elysium Mons.

Methods:

L 490: A substantial amount of the analysis from here onward appears to rely on the assumption that seismic noise does not follow a BL distribution due to the decreased dynamic range. How carefully has this assumption been tested? For example, if the data are filtered in the 0.2-0.9Hz range as per other papers, does this noticeably change the BL distribution of noise?

We have carefully tested the BL behaviour through independent work over the past years. When the dynamic range tends to infinity, the distribution of deviates perfectly follows Benford’s Law; when the dynamic range approaches zero, their distribution does not obey Benford’s Law (see, for example, Sambridge et al., 2011). Our analysis here reveals that the BL method is susceptible to large-amplitude glitches in the Martian waveform data. Therefore, it is concluded that the BL cannot be used to detect the aftershocks or repeated noise-masked marsquakes. We mention this in the Materials and Methods section’s paragraph in lines 492-496.

Sambridge, M., H. Tkalčić, and P. Arroucau, Benford's Law of First Digits: From Mathematical Curiosity to Change Detector, Asia Pacific Mathematics Newsletter, 1, No. 4, 1-5, 2011.

L 496: would it be fairer (and clearer) to instead phrase this as ‘the ground motion time series during the two marsquakes analysed follows a BL distribution’? Because ‘confirm’ suggests that they were known a priori to be BL distributed, which as far as I know is not known for Mars.

OK. We rephrase the sentence as you suggested, i.e., “assist the detection”.

L 529: is this true for all events or just 0173a? This is confusing.

This was explicitly stated for S0173a in Clinton et al. (2021). But it is also true for S0235b, as observed from its spectrogram in Figure 14 of Clinton et al. (2021).

Clinton, J. F., Ceylan, S., van Driel, M., Giardini, D., Stähler, S. C., Böse, M., . . . Stott, A. E. (2021). The Marsquake catalogue from InSight, sols 0–478. Physics of the Earth and Planetary Interiors, 310, 106595. doi:10.1016/j.pepi.2020.106595

As per the response to the first reviewer, it is not clear to me how the authors can be sure that the strong seismic noise is not contaminating the MF results? Especially given the strong diurnal variation in this noise, I would expect some influence upon the likelihood of matching a template event which occurred during the quiet part of the day (as the largest quakes are) to the noisiest part of the day where new events are identified.

We have considered these concerns and conducted comprehensive tests included in Section S2. More specifically, Section S2.4, ‘Tests of events buried in continuous noise’, consists of the tests on synthetic noise. These tests confirm the robustness of the MF results.

We also check the maximum raw amplitudes of the S arrivals for all the S0173a and S0235b detections listed in Table 1 for all three components (see the figure appended below). The red and blue dotted lines denote the maximum amplitudes of the S arrival for S0173a and S0235b, while the black dotted lines display the mean value of the maximum amplitude of the S arrival for all MF detections. As it can be seen, these MF detections have comparable amplitudes with S0173a and S0235b. Moreover, the maximum amplitudes of the detections are around 1,000 units or less on the vertical axis, which is considerably smaller than the amplitudes of the wind noise reaching 20,000 or more, as shown in Figure S2.1. This illustrates that the MF detections are located in the quiet diurnal time, and the noise shows substantial diurnal variations, as you point out.

Overall comments:

This manuscript is now more clearly presented, and I at least understand the basis of the methodology that the authors are applying. I must admit that I am still unconvinced about the results, however - not least because there are number of questions I had that are still

unresolved. For example (summarising some and repeating from my previous comments, in no particular order):

What is the impact of the noise on the MF method, and how is this robustly explored and quantified?

We have put substantial effort into the previous revision to address these concerns, which has been recognized by the additional reviewer and through some of your previous comments. To address the impact of noise on the MF method, we embedded the S0173a and S0235b in the synthetic noise, as shown in Section S2.4, to rigorously test the influence of noise. The shown ratios of signal to noise are 1.0, 0.75, and 0.5. From these tests, we can conclude that the MF method works well for a moderate noise but cannot identify weak marsquakes or those significantly contaminated by ambient noise. Unless an alternative method is found in the future, those marsquakes will stay undetected. Reflecting our thoughts on the conditions on Earth, many earthquakes still go undetected due to the imperfections of our techniques and limited station coverage.

What is the authors' interpretation of the diurnal patterning (or lack thereof) in matched events? How does this compare with previous papers, and how is this discrepancy resolved with a satisfactory explanation?

As illustrated in Table 1 and Figure 3, the MF detections point to the marsquakes occurring daily and nightly, thus exhibiting a pattern not different from earthquakes. The simplest explanation is that the insofar known marsquakes are picked only during the quiet night by the InSight team due to the strong wind noise present during the day. Owing to the capability of the MF method to enhance the detections, we hope the smaller marsquakes will supplement the InSight marsquake catalog from now on.

Does BL actually 'detect hitherto undetected events' (as per the abstract), or is it only used to 'confirm the main low-frequency effects (in the methods section)? If it is only the latter, I wonder whether it needs to be included in the paper in such a detailed manner? The space might be better used exploring the questions raised above about the MF method.

We agree that Benford's Law (BL) does not have to be included in the paper in a detailed matter. There are no figures related to the BL in the main text, and in most instances, it is mentioned in passing. We attempted to move most information on the method to the supplements as it can only be used to assist the main low-frequency events at this stage.

My previous comments about it being surprising that some events are repeating whilst others are not still stands: this does not necessarily mean that there is a mistake in the workflow, but can any reasonable suggestion or speculation be made? Of course, not every method will work for every event, but without understanding why this is it is harder to accept that this method is reliable and robust. I do not quite follow the suggestion about 'suppression' of wind noise.

Earthquakes are strongly non-linear systems. On Earth, some earthquakes are repetitive, and some are not. There are specific regions where repetitive earthquakes exist and are well documented in the near-surface and the shallow to Earth's deep structures' investigations, e.g., from the applications in studying glacial quakes and measuring the crustal stress before

and after the occurrence of large earthquakes, to the applications for the differential rotation of the inner core relative to the Earth's mantle. Why some quakes are repetitive, and others are not is probably driven by the lithospheric stress field and underground architecture itself. This may offer a clue for why some of the observed marsquakes are repetitive, and others are not. On a more subtle level, there might be events that fall behind our detection capacity, even with enhanced tools such as the MF method.

As for the methods themselves, the success of the MF method stems from the waveform similarities between the template and the detected events. This implies the proximity of their location and similar focal mechanisms, which then produce highly similar waveforms.

As in Table 1, most of the newly detected events (45 out of 47) are related to the S0173a and S0235b events, showing clear P- and S-wave onsets. The other two detections are associated with the two B-quality events: S0183a and S0325a. The failure to detect other events may suggest the tectonic heterogeneities on Mars.

The supplied code is not in a usable format - I realise that this is not formally to be reviewed, but I opened it to try and follow the workflow and the README was not sufficient to know what was going on.

Thanks for pointing this out. We tried to improve the quality of the README file further.

In summary, we much appreciate your valuable time and constructive criticism, which helped us improve the quality of the manuscript substantially.

Reviewer #4 (Remarks to the Author):

Using the Matched Filter method and Benford's Law, this study attempted to detect previously undetected events from seismic data recorded by the InSight seismometer SEIS. The authors reported 47 new seismic events, most of which are attributed to repetitive quakes with two previously detected events. Given the low signal-to-noise ratio of the seismic data, the authors' effort to extract useful signals from the raw data is very valuable and promising. The techniques used in this study may have significant potential in applying to InSight seismic data and other planetary seismic data in future. Also, if the newly detected events are robust and reliable, the new results have significant implications for the internal dynamics and evolution of Mars.

The major concern raised by other reviewers for the original manuscript is the reliability and robustness of the methods and results. In the revised manuscript, the authors have made comprehensive tests to reinforce the reliability and robustness of the results. Therefore, I would recommend the paper can be accepted for publication.

Thank you very much for your supportive comments.

I only have some minor suggestions:

Line32: "astronomical considerations" should be more specific. I think the major constraint in ref(7) is tidal response, i.e. Love number k_2 .

We changed ‘astronomical’ to ‘geodetic’.

Line50: Discussion of the Martian dynamo and core state should refer to “Stevenson, D. J. (2001). Mars’ core and magnetism. *Nature*, 412(6843), 214–219.”

We refer to the literature as you suggested.

Line51: Again, “the satellite data” should be more specific here.

We changed it to ‘the Love number inferred from Mars Global Surveyor tracking data’ (Lines 49-50).

Line98: Although two methods are detailed in the Method section, it would be helpful to briefly describe the principle of the methods in the introduction.

We now mention the two methods in the last paragraph of the introduction section (Lines 95-98).

Line103: “the same general location” is a bit vague to me.

Changed to ‘assuming they are collocated’ (Line 101). In seismology, the term co-location indicates that the two locations are within the characteristic wavelength at which the observations are made.

REVIEWER COMMENTS

Reviewer #3 (Remarks to the Author):

Thank you for your response to the comments made. Whilst I am still unsure about some of the conclusions, I believe that the methodology is now thoroughly enough explained that this could be published for others to consider.